# The prevalence of SARS-CoV-2 infection and other public health outcomes during the BA.2/BA.2.12.1 surge, New York City, April–May 2022

Saba A. Qasmieh[1,2], McKaylee M. Robertson [1,2], Chloe A. Teasdale[1,2], Sarah G. Kulkarni[1], Heidi E. Jones[1,2], David A. Larsen [3], John J. Dennehy [4], Margaret McNairy[1,5], Luisa N. Borrell [2] & Denis Nash [1,2✉]

## Abstract

**Background** Routine case surveillance data for SARS-CoV-2 are incomplete, unrepresentative, missing key variables of interest, and may be increasingly unreliable for timely surge detection and understanding the true burden of infection.

**Methods** We conducted a cross-sectional survey of a representative sample of 1030 New York City (NYC) adult residents ≥18 years on May 7-8, 2022. We estimated the prevalence of SARS-CoV-2 infection during the preceding 14-day period. Respondents were asked about SARS-CoV-2 testing, testing outcomes, COVID-like symptoms, and contact with SARS-CoV-2 cases. SARS-CoV-2 prevalence estimates were age- and sex-adjusted to the 2020 U.S. population. We triangulated survey-based prevalence estimates with contemporaneous official SARS-CoV-2 counts of cases, hospitalizations, and deaths, as well as SARS-CoV-2 wastewater concentrations.

**Results** We show that 22.1% (95% CI 17.9–26.2%) of respondents had SARS-CoV-2 infection during the two-week study period, corresponding to ~1.5 million adults (95% CI 1.3-1.8 million). The official SARS-CoV-2 case count during the study period is 51,218. Prevalence is estimated at 36.6% (95% CI 28.3–45.8%) among individuals with co-morbidities, 13.7% (95% CI 10.4–17.9%) among those 65+ years, and 15.3% (95% CI 9.6–23.5%) among unvaccinated persons. Among individuals with a SARS-CoV-2 infection, hybrid immunity (history of both vaccination and infection) is 66.2% (95% CI 55.7–76.7%), 44.1% (95% CI 33.0–55.1%) were aware of the antiviral nirmatrelvir/ritonavir, and 15.1% (95% CI 7.1–23.1%) reported receiving it. Hospitalizations, deaths and SARS-CoV-2 virus concentrations in wastewater remained well below that during the BA.1 surge.

**Conclusions** Our findings suggest that the true magnitude of NYC's BA.2/BA.2.12.1 surge may have been vastly underestimated by routine case counts and wastewater surveillance. Hybrid immunity, bolstered by the recent BA.1 surge, likely limited the severity of the BA.2/ BA.2.12.1 surge.

## Plain language summary

It is difficult to assess the true prevalence of SARS-CoV-2, the virus that causes COVID-19, due to changes in testing practices and behaviors, including increasing at-home testing and decreasing healthcare provider-based testing. We conducted a population-representative survey in New York City to estimate the prevalence of SARS-CoV-2 during the second Omicron surge in spring 2022. We compared survey-based SARS-CoV-2 prevalence estimates with data on diagnosed cases, hospitalizations, deaths, and SARS-CoV-2 concentration in wastewater. Our survey-based estimates were nearly 30 times higher than official case counts and estimates of immunity among those with active infection were high. Taken together, our results suggest that the magnitude of the second Omicron surge was likely significantly underestimated, and high levels of immunity likely prevented a major surge in hospitalizations/deaths. Our findings might inform future work on COVID-19 surveillance and how to mitigate its spread.

[1] Institute for Implementation Science in Population Health (ISPH), City University of New York (CUNY), New York, NY, USA. [2] Department of Epidemiology and Biostatistics, Graduate School of Public Health and Health Policy, City University of New York (CUNY), New York, NY, USA. [3] Department of Public Health, Falk College, Syracuse University, Syracuse, NY, USA. [4] Department of Biology, Queens College, City University of New York, Queens, NY, USA. [5] Center for Global Health and Division of General Internal Medicine, Weill Cornell Medicine, New York, NY, USA. ✉email: denis.nash@sph.cuny.edu

Major surges in SARS-CoV-2 transmission, due to new or evolving variants and waning population immunity, are expected in many parts of the world for the foreseeable future. Depending on variant properties and the timing of surges in relation to population immunity, the impact of surges could be severe even in highly vaccinated populations. The Omicron (BA.1) surge in the U.S., beginning mid-December 2021 when 62% of the U.S. population was fully vaccinated, overwhelmed the health care system and resulted in more than 187,000 deaths during a 4-month period[1,2]. In the current phase of the pandemic, key components of the U.S. strategy to limit the impact of SARS-CoV-2 surges are vaccinations, timely boosters and, for those most vulnerable, prophylaxis with monoclonal antibodies, and rapid treatment with oral antivirals or monoclonal antibodies, which can greatly reduce the risk of severe disease and death (i.e., secondary prevention)[3–5]. As the pandemic progresses, levels of hospitalization and death among those most vulnerable to severe COVID-19 are likely to vary by locality, and to be influenced by variant properties (transmissibility, severity, immune evasion), population levels of immune protection (via vaccination/boosters and/or prior infection), varying intervals between surges, which in the case of longer intervals, protection may wane and leave some communities and sub-populations more susceptible to surges in hospitalizations and deaths, and access to treatment (antivirals, monoclonal antibodies).

The complex and evolving nature of the U.S. pandemic has led to calls for more robust, timely, and representative approaches to public health surveillance[6,7]. Routine passive surveillance to inform the public health response relies on healthcare providers, testing providers, and laboratories to report data on those who are tested. Although these surveillance data have been essential for tracking and responding to the COVID-19 pandemic, routinely reported testing data have become increasingly unreliable for timely surge detection and gauging the overall surge magnitude and sub-population burden[6–9]. For example, during the latter half of NYC's initial Omicron surge (BA.1), official case counts were likely 3–4 times lower than an estimate of infections from a representative sample of the adult population[10]. Data from traditional (passive) surveillance underestimate the true burden of infection in the general population due to undiagnosed/untested cases[11], underreporting of cases by providers and labs, as well as the expanding use of at-home rapid antigen tests, which are not reflected in routine case surveillance in the U.S[8,10,12]. Moreover, while SARS-CoV-2 provider and laboratory reporting is believed to be incomplete[6], the extent of incomplete reporting has not been systematically evaluated in NYC or nationally, and may be influenced by surges in transmission, testing demand, or both. The degree of underdiagnosis and underreporting is likely differential by geographic and sociodemographic factors and variable over time[13,14], and may prevent or delay surge detection, limiting the ability of individuals and governments to take precautions.

Surveillance data are also limited with regard to key information about cases such as race/ethnicity, vaccination status, history of prior SARS-CoV-2 infection, comorbidities and uptake of biomedical interventions such as oral antivirals and monoclonal antibodies. The lack of such information prevents systematic assessment of both the burden of infection and uptake of biomedical interventions among those who may be most vulnerable to a severe outcome. Population-based surveys have been used as part of routine public health surveillance in the United Kingdom[15] and NYC[16] and can provide rapid and complementary information that addresses some of the limitations of traditional surveillance of SARS-CoV-2 cases, hospitalizations, and deaths[6,7].

This study aimed to use an efficient and pragmatic survey-based approach to assess the burden of SARS-CoV-2 infection during the BA.2/BA.12.2.1 surge in NYC starting in March 2022. In addition to prevalence data, the survey captured data on clinical outcomes, and the intersection of vaccine-induced and infection-induced immunity. We triangulated information from our survey with official counts of cases, hospitalizations, and deaths from routine surveillance and with data on SARS-CoV-2 concentration in NYC wastewater for the same time period.

We report on a population-representative survey of adults in NYC to estimate the prevalence of SARS-CoV-2 infection during the BA.2/BA.12.2.1 Omicron surge in late April/early May 2022. We compare survey-based SARS-CoV-2 prevalence estimates with city-wide SARS-CoV-2 data on diagnosed cases, hospitalizations, deaths, and SARS-CoV-2 concentration in wastewater. We found that survey-based prevalence estimates were nearly 30 times higher than official case counts, and estimates of recently acquired hybrid immunity among those with active infection were high. We conclude that no single data source provides a complete or accurate assessment of the epidemiologic situation. Taken together, however, our results suggest that the magnitude of the BA.2/BA.12.2.1 Omicron surge was likely significantly underestimated, and high levels of hybrid immunity likely prevented a major surge in hospitalizations/deaths.

## Methods

**Survey-based estimation of SARS-CoV-2 prevalence.** We conducted a cross-sectional survey, in English and Spanish, during May 7–8, 2022, of 1030 adult NYC residents via landlines (IVR) and mobile phones (SMS text). Potential participants were randomly selected from a sampling frame. Additional details on the survey design and sampling are provided as Supplementary Methods. Respondents were asked about SARS-CoV-2 testing and related outcomes during the 14 days prior to the survey (April 23–May 8). During the same time period, the BA.2.12.1 subvariant rose from an estimated 32% of reported cases to 47%[17].

The study protocol was approved by the Institutional Review Board at the City University of New York (CUNY) (approval number 2022-0131-PHHP, with a waiver for written informed consent given deidentification of data and minimal risk to study subjects).

*Point prevalence estimation.* The survey questionnaire (Supplementary Note 1) ascertained SARS-CoV-2 testing, including the location, types and results of viral diagnostic tests taken in the 14 days prior to the survey (PCR, rapid antigen and/or at-home rapid tests). The survey also captured information on COVID-19 symptoms, as well as known close contacts with a confirmed or probable case of SARS-CoV-2 infection. COVID-19 symptoms included any of the following: fever of ≥100°F, cough, runny nose and/or nasal congestion, shortness of breath, sore throat, fatigue, muscle/body aches, headaches, loss of smell/taste, nausea, vomiting and/or diarrhea[18]. Participants were also asked about vaccination status, comorbidities that increase vulnerability to severe COVID-19, and prior history of SARS-CoV-2 infection/COVID. Participants who reported any type of COVID-19 test with a healthcare or testing provider, regardless of the result, were asked about awareness and uptake of the antiviral nirmatrelvir/ritonavir oral tablets (Paxlovid™), which NY State made available in the Spring of 2022.

Information gathered from respondents was used to estimate the number and proportion of respondents who likely had SARS-CoV-2 infection during the study period based on the following mutually exclusive, hierarchical case classification: (1) Confirmed case: self-report of one or more positive tests with a health care or testing provider; or (2) Probable case: self-report of a positive test

result exclusively on at-home rapid tests (i.e. those that were not followed up with confirmatory diagnostic testing with a provider); or (3) Possible case: self-report of COVID-like symptoms AND a known epidemiologic link (close contact) to one or more laboratory confirmed or probable (symptomatic) SARS-CoV-2 case(s)[18] in a respondent who reported never testing or only testing negative during the study period.

The intersection of vaccine- and infection-induced immunity: We combined information on vaccination status with that on prior COVID infections. Those who were fully vaccinated and those who were also boosted (fully vaccinated/boosted) with a history of prior COVID were classified as having 'hybrid immunity' against severe COVID-19; those who were fully vaccinated or boosted with no history of prior COVID were classified as having 'vaccine-induced immunity only'; those who were not fully vaccinated but had a history or prior COVID were classified as having 'infection-induced immunity only'; and those who were neither vaccinated/boosted nor had a history of COVID were classified as having 'no prior immunity' (SARS-CoV-2 näive).

Statistical analysis: We estimated the prevalence of SARS-CoV-2 by socio-demographic characteristics, NYC borough (county), vaccination status, comorbidity and prior SARS-CoV-2 infection status. Survey weights were applied to generate weighted numbers and estimates of the proportion who had active SARS-CoV-2 infection at any time during the study period along with 95% confidence interval (95% CI). We applied these weighted sample proportions and 95% CI to the 6,740,580 NYC residents ≥18 years to obtain estimates of the absolute number of NYC adults with SARS-CoV-2 infection[19]. We used direct standardization to present age- and sex-adjusted prevalence estimates using the U.S. 2020 census. Crude and age- and sex-adjusted prevalence ratios were estimated with a log-binomial model. Pearson's $\chi^2$ test was performed to assess associations between each factor and testing status. Analyses were conducted using SAS and R.

**SARS-CoV-2 routine testing and case surveillance data**. We used publicly available, daily aggregated data on the number of SARS-CoV-2 tests, test types (PCR or rapid antigen), and results through 10 June 2022 to describe the number of tests and positive tests with health care providers and testing providers reported to the NYC Department of Health and Mental Hygiene(DOHMH) during the study period[20].

**SARS-CoV-2 wastewater surveillance data**. We analyzed publicly available data on SARS-CoV-2 concentrations in NYC wastewater through June 5, 2022, which is estimated based on weekly influent samples from 14 water resource recovery facilities (WRRFs) in NYC covering wastewater of an estimated 8.2 million residents[21]. Specifically, WRRFs are sampled up to twice each week and per capita SARS-CoV-2 load (N1 copies per capita) is reported for each sample date[21]. We plotted the mean per capita SARS-CoV-2 load (N1 copies per capita) by sample date across all 14 WRRFs. Details on sampling and laboratory methods and measurement are available in the public use dataset documentation[21].

**Reporting summary**. Further information on research design is available in the Nature Portfolio Reporting Summary linked to this article.

## Results

**Survey**. An estimated 22.1% (95% CI 17.9–26.2%) of 1030 respondents had SARS-CoV-2 infection in the 14 days prior to the interview, corresponding to 1.5 million adults (95% CI 1.3–1.8 million) (Table 1). The estimate of 22.1% includes: (1) 11.4% (95% CI 8.4–14.3%) who were positive based on one or more tests with a health care or testing provider (confirmed cases); (2) 6.5% (95% CI 4.2–8.8%) who were positive exclusively based on one or more positive at-home rapid tests (probable cases); and (3) 4.2% (95% CI 1.8–6.7%) who met the definition for possible SARS-CoV-2 infection based on having *both* COVID-like symptoms and a close contact with a confirmed/probable case. About 53.8% of adults in our survey reported having any SARS-CoV-2 test during the study period, including 43% who reported testing with a health care or testing provider (5.5% exclusively) and 48% who tested using an at-home rapid test (10.9% exclusively).

The weighted characteristics of survey participants and period prevalence estimates (both crude and age/sex-adjusted) are also shown in Table 1. In general, crude prevalence estimates and prevalence ratios were not materially altered by age and sex adjustment. SARS-CoV-2 prevalence was high among all groups, but varied substantially by sociodemographic factors, and was especially high among adults aged 18–24 (26.1%, 95% CI 14.2–42.9%) and 45–54 (28.0%, 95% CI 17.7–41.2%). Age- and sex-adjusted prevalence was higher among Hispanic (31.1%, 95% CI 22.6–41.1%), and non-Hispanic white residents (26.0%, 95% CI 20.5–32.4%), and those with some high school education or less (31.3%, 95% CI 20.8–44.2%). Age- and sex-adjusted prevalence estimates were the lowest among non-Hispanic Black (11.4%, 95% CI 6.7–18.7%) and Asian/Pacific Islander (5.2%, 95% CI 1.7–14.8%) residents. Age- and sex-adjusted SARS-CoV-2 prevalence among survey respondents increased in dose response fashion with the number of household members (15.0%, 95% CI 9.8–22.6% vs 21.1%, 95% CI 15.9–27.5% vs 25.7%, 95% CI 17.9–35.3%), and households with children <18 years had substantially higher prevalence than in those households without children (31.5%, 95% CI 12.9–43.0% vs 17.8%, 95% CI 13.6–22.8%; Fig. 1).

Individuals who were fully vaccinated with a booster had higher age- and sex-adjusted SARS-CoV-2 prevalence (25.2%, 95% CI 20.0–31.3%) than those who were fully vaccinated but not boosted (11.8%, 95% CI 5.7–23.0%) and those who were unvaccinated (15.3%, 95% CI 9.6–23.5%; Table 2). Those who said they tested positive for SARS-CoV-2 once before the current episode (36.9%, 95% CI 27.6–47.3%) or more than once (37.5%, 95% CI 27.2–49.1%) had much higher age- and sex-adjusted prevalence than those who said they never tested positive before (11.5%, 95% CI 6.7–18.9%) or who thought they had COVID before but never tested positive (13.0%, 95% CI 8.2–20.0%).

*Hybrid immunity*. Among those who were either vaccinated/boosted, those who also had a SARS-CoV-2 infection in the past (hybrid immunity) had an age- and sex-adjusted prevalence of 29.2% (95% CI 23.1–36.0%), compared with 12.9% (95% CI 7.3–21.3%) among those who did not have SARS-CoV-2 in the past (vaccine-induced protection only; Table 2). Among those who were not vaccinated/boosted, those who had SARS-CoV-2 infection in the past (infection-induced immunity only) had an age- and sex-adjusted prevalence of 24.6% (95% CI 16.4–35.2%), compared with 1.7% (95% CI 0.3–10.7%) among those who did not have a SARS-CoV-2 infection in the past (no prior SARS-CoV-2 immunity). The proportion of adults with hybrid immunity and infection-induced immunity only were higher in those with a SARS-CoV-2 infection versus those without (Fig. 2).

*Vulnerability to severe COVID-19*. The estimated age- and sex-adjusted prevalence of SARS-CoV-2 was substantial among those groups who are more vulnerable to severe SARS-CoV-2 and death, including unvaccinated persons (15.3%, 95% CI 9.6–23.5%), those aged 65+ (13.7%, 95% CI 10.4–17.9%), and

**Table 1 Characteristics of survey respondents by testing status and prevalence of SARS-CoV-2, NYC April–May 2022**

| | Total<br>Weighted N (%) | Crude prevalence of SARS-CoV-2 infection[a]<br>% (95% CI) | Standardized prevalence of SARS-CoV-2 infection[b]<br>% (95% CI)<br>(Weighted N = 991) | Crude prevalence ratio (PR)<br>PR (95% CI)<br>(Weighted N = 1030) | Adjusted prevalence ratio (aPR)[c]<br>aPR (95% CI)<br>(Weighted N = 1030) |
|---|---|---|---|---|---|
| Total | 1030 (100) | 22.1 (17.9–26.2) | – | – | – |
| **Age** | | | | | |
| 18–24 | 112 (10.8) | 27.7 (12.7–42.8) | 26.1 (14.2–42.9) | 1.3 (0.9–1.9) | 1.4 (0.9–2.00 |
| 25–34 | 232 (22.5) | 21.6 (12.0–31.1) | 21.4 (13.4–32.4) | -ref- | -ref- |
| 35–44 | 176 (17.1) | 20.5 (8.1–32.8) | 21.1 (11.0–36.6) | 0.9 (0.6–1.4) | 1.0 (0.7–1.4) |
| 45–54 | 159 (15.5) | 27.8 (15.9–39.7) | 28.0 (17.7–41.2) | 1.3 (0.9–1.8) | 1.3 (0.9–1.9) |
| 55–64 | 158 (15.3) | 23.6 (15.4–31.9) | 22.4 (15.3–31.6) | 1.1 (0.8–1.6) | 1.1 (0.8–1.6) |
| 65+ | 194 (18.8) | 14.9 (11.0–18.8) | 13.7 (10.4–17.9) | 0.7 (0.5–1.0) | 0.7 (0.5–1.1) |
| **Gender** | | | | | |
| Male | 471 (45.7) | 23.4 (16.5–30.2) | 22.8 (17.0–30.0) | 1.2 (0.9–1.5) | 1.1 (0.9–1.5) |
| Female | 527 (51.2) | 20.1 (14.9–25.3) | 20.1 (15.4–25.8) | -ref- | -ref- |
| Non-binary[d] | 32 (3.1) | 36.1 (13.2–59.1) | 29.1 (15.2–48.6) | 1.8 (1.1–2.9) | 1.9 (1.1–3.1) |
| **Race/Ethnicity** | | | | | |
| Black NH | 214 (20.8) | 11.3 (4.9–17.6) | 11.4 (6.7–18.7) | 0.4 (0.3–0.6) | 0.4 (0.3–0.6) |
| White NH | 393 (38.2) | 26.3 (20.0–32.5) | 26.0 (20.5–32.4) | -ref- | -ref- |
| Hispanic | 258 (25.0) | 33.6 (23.4–43.9) | 31.1 (22.6–41.1) | 1.3 (1.0–1.6) | 1.3 (1.0–1.7) |
| Asian/Pacific Isl. | 121 (11.7) | 4.9 (0.0–10.5) | 5.2 (1.7–14.8) | 0.2 (0.1–0.4) | 0.2 (0.1–0.4) |
| Other | 45 (4.3) | 17.1 (0.0–35.9) | 14.1 (5.3–32.6) | 0.7 (0.3–1.3) | 0.6 (0.3–1.2) |
| **Years of education** | | | | | |
| Some HS and below | 164 (16.0) | 28.8 (15.8–41.8) | 31.3 (20.8–44.2) | 1.3 (0.9–1.9) | 1.4 (1.0–2.1) |
| HS graduate | 239 (23.2) | 21.5 (12.3–30.8) | 20.0 (12.2–30.9) | -ref- | -ref- |
| Some college | 205 (19.9) | 16.7 (9.6–23.7) | 16.3 (10.6–24.5) | 0.8 (0.5–1.1) | 0.8 (0.5–1.2) |
| ≥College graduate | 422 (40.9) | 22.4 (16.3–28.5) | 23.1 (17.8–29.5) | 1.0 (0.8–1.4) | 1.0 (0.8–1.4) |
| **Household size** | | | | | |
| 1 | 349 (33.8) | 16.7 (10.4–23.0) | 15.0 (9.8–22.6) | -ref- | -ref- |
| 2–3 | 403 (39.1) | 20.7 (14.8–26.7) | 21.1 (15.9–27.5) | 1.2 (0.9–1.7) | 1.2 (0.8–1.6) |
| 4+ | 278 (27.1) | 30.8 (20.9–40.6) | 25.7 (17.9–35.3) | 1.8 (1.4–2.5) | 1.7 (1.3–2.4) |
| **Any children <18 yrs** | | | | | |
| Yes | 271 (26.3) | 33.3 (23.0–43.5) | 31.5 (12.9–43.0) | 1.8 (1.5–2.3) | 1.7 (1.5–2.4) |
| No | 759 (73.7) | 18.1 (13.8–22.3) | 17.8 (13.6–22.8) | -ref- | -ref- |
| **Household income** | | | | | |
| Below 25 K | 221 (21.5) | 18.6 (10.0–27.2) | 14.2 (8.2–23.5) | 0.6 (0.4–0.8) | 0.6 (0.4–0.8) |
| 25,000–65,000 | 287 (27.9) | 24.9 (16.7–33.0) | 27.5 (19.3–37.5) | 0.8 (0.6–1.0) | 0.7 (0.6–0.9) |
| 65,000–150,000 | 229 (22.2) | 31.4 (20.8–42.1) | 32.3 (24.1–41.7) | -ref- | -ref- |
| Above 150,000 | 77 (7.5) | 17.9 (7.2–28.6) | 18.6 (10.5–30.8) | 0.6 (0.3–1.0) | 0.5 (0.3–0.9) |
| Prefer not to answer | 215 (20.9) | 13.5 (6.7–20.2) | 14.2 (7.8–24.4) | 0.4 (0.3–0.6) | 0.4 (0.3–0.6) |
| **Employed** | | | | | |
| Yes | 477 (46.3) | 29.3 (22.2–36.5) | 30.9 (24.8–37.9) | -ref- | -ref- |
| No/DK | 553 (53.7) | 15.8 (11.4–20.3) | 13.4 (8.8–19.9) | 0.5 (0.4–0.7) | 0.5 (0.4–0.7) |

[a]Estimates are weighted to the US adult population.
[b]Direct standardized for the age and sex groupings in the 2020 U.S. census.
[c]Model adjusted for sex and age.
[d]Estimates are adjusted for age only

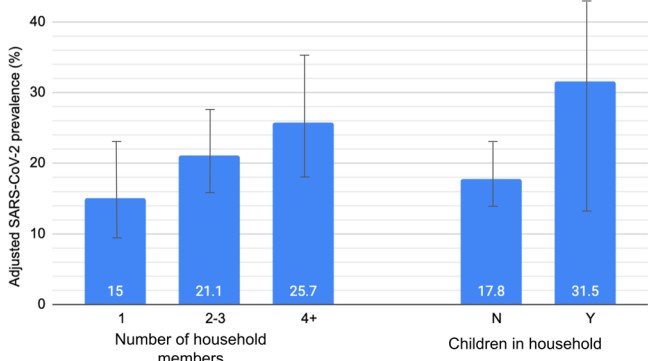

**Fig. 1 Associations of household size and children in the household with SARS-CoV-2 prevalence.** Age- and sex-adjusted SARS-CoV-2 prevalence estimates and 95% confidence interval (95% CI) among NYC adults by household size and presence of children in the household, April–May, 2022.

individuals with co-morbidities (36.6%, 95% CI 28.3–45.8%; Table 2). Among those with any of these vulnerabilities to severe COVID-19 (age≥65, comorbidities, unvaccinated), the age- and sex-adjusted prevalence of SARS-CoV-2 infection was 29.6%

(95% CI 23.7–36.3%). Also among this vulnerable group, only 68.8% (95% CI 62.8–74.8%) were vaccinated/boosted. Specifically, 60.0% (95% CI 59.9–66.2%) were fully vaccinated *and* boosted (of whom 68.8% [95% CI 61.5–76.2%] had a history of prior COVID) and 8.7% (95% CI 5.6–11.8%) were fully vaccinated but not boosted (of whom 71.8% [95% CI 54.4–89.1%] had a history of prior COVID). However, 31.2% (95% CI 25.2–37.2%) were unvaccinated (of whom 62.1% [95% CI 50.1–74.1%] had a history of prior COVID).

*Testing.* Just over half of NYC adults (53.8%) reported any SARS-CoV-2 testing during the study period, including at-home testing. A substantial proportion of those testing reported doing so with a provider (42.9%), corresponding to an estimated 2.9 million provider tests (Supplementary Data 1). Additionally, 10.9% of adults said they tested only at home. Compared with those who did not test during the study period, those who tested were more likely to be younger, Hispanic, and non-Hispanic white, and to have higher education, larger households, children in the household, lower household income, received a booster, had COVID more than once, hybrid and infection-induced immunity, medical vulnerabilities to severe COVID-19, and no insurance (Supplementary Data 1).

**Table 2 Additional characteristics of survey respondents by testing status and prevalence of SARS-CoV-2, NYC April–May 2022**

| | Total | Crude prevalence of SARS-CoV-2 infection[a] | Standardized prevalence of SARS-CoV-2 infection[b] | Crude prevalence ratio (PR) | Adjusted prevalence ratio (aPR)[c] |
|---|---|---|---|---|---|
| | Weighted N (%) | % (95% CI) | % (95% CI) (Weighted N = 991) | PR (95% CI) (Weighted N = 1030) | aPR (95% CI) (Weighted N = 1030) |
| Total | 1030 (100) | 22.1 (17.9–26.2) | – | – | – |
| Nativity | | | | | |
| Born in U.S. | 714 (69.4) | 26.8 (21.3–32.2) | 26.5 (21.6–32.2) | -ref- | -ref- |
| Born outside U.S. | 316 (30.6) | 11.4 (6.6–16.2) | 10.8 (6.5–17.3) | 0.4 (0.3–0.6) | 0.4 (0.3–0.6) |
| NYC borough | | | | | |
| Bronx | 175 (17.0) | 21.1 (10.4–31.8) | 20.2 (12.0–32.0) | 0.9 (0.6–1.4) | 0.9 (0.6–1.4) |
| Brooklyn | 317 (30.7) | 27.2 (19.8–34.7) | 26.1 (19.7–33.8) | 1.2 (0.9–1.6) | 1.2 (0.9–1.7) |
| Manhattan | 201 (19.5) | 22.6 (13.2–32.1) | 24.0 (15.5–35.1) | -ref- | -ref- |
| Queens | 278 (27.0) | 18.2 (10.4–26.1) | 17.6 (11.3–26.3) | 0.8 (0.6–1.2) | 0.8 (0.6–1.2) |
| Staten Island | 59 (5.7) | 13.6 (0.0–28.0) | 8.1 (2.8–20.9) | 0.6 (0.3–1.2) | 0.6 (0.3–1.3) |
| Vaccination status | | | | | |
| Boosted | 691 (67.1) | 25.2 (19.8–30.5) | 25.2 (20.0–31.3) | 2.1 (1.4–3.4) | 2.1 (1.3–3.2) |
| Fully vaccinated not boosted | 149 (14.5) | 11.8 (3.5–20.1) | 11.8 (5.7–23.0) | -ref- | -ref- |
| Not vaccinated | 190 (18.5) | 18.9 (10.2–27.5) | 15.3 (9.6–23.5) | 1.6 (0.9–2.7) | 1.5 (0.9–2.5) |
| Prior COVID | | | | | |
| Never | 390 (37.9) | 10.7 (4.7–16.6) | 11.5 (6.7–18.9) | 0.3 (0.2–0.4) | 0.4 (0.4–0.6) |
| Once | 235 (22.8) | 39.2 (28.2–50.2) | 36.9 (27.6–47.3) | -ref- | -ref- |
| More than once | 140 (13.6) | 40.4 (27.2–53.6) | 37.5 (27.2–49.1) | 1.0 (0.8–1.3) | 1.0 (0.8–1.3) |
| No positive test, but likely COVID | 265 (25.8) | 14.1 (8.5–19.7) | 13.0 (8.2–20.0) | 0.4 (0.3–0.5) | 0.5 (0.4–0.7) |
| Protection against severe disease | | | | | |
| Hybrid protection | 522 (50.6) | 28.9 (22.6–35.1) | 29.2 (23.1–36.0) | -ref- | -ref- |
| Vaccine-induced protection only | 318 (30.9) | 12.9 (5.6–20.1) | 12.9 (7.3–21.3) | 0.4 (0.3–0.6) | 0.4 (0.3–0.6) |
| Infection-induced protection only | 118 (11.5) | 29.8 (16.6–43.1) | 24.6 (16.4–35.2) | 1.0 (0.8–1.4) | 0.9 (0.7–1.3) |
| No protection | 72 (7.0) | 0.9 (0.0–2.7) | 1.7 (0.3–10.7) | 0.0 (0.0–0.4) | 0.0 (0.0–0.3) |
| Comorbidities | | | | | |
| Yes | 377 (36.6) | 34.9 (26.9– 42.8) | 36.6 (28.3–45.8) | 2.4 (1.9–3.0) | 2.8 (2.2–3.5) |
| No | 653 (63.4) | 14.7 (10.4–19.0) | 14.7 (11.1–19.1) | -ref- | -ref- |
| Any vulnerability[d] | | | | | |
| Yes | 609 (59.1) | 27.8 (22.1–33.5) | 29.6 (23.7–36.3) | 2.0 (1.5–2.7) | 2.5 (1.9–3.3) |
| No | 421 (40.9) | 13.8 (8.3–19.2) | 16.7 (11.4–23.7) | -ref- | -ref- |
| Health insurance | | | | | |
| Yes | 836 (81.2) | 22.9 (18.2–27.6) | 22.6 (18.2–27.8) | -ref- | -ref- |
| No | 194 (18.9) | 18.5 (9.9–27.0) | 17.3 (10.6–27.0) | 0.8 (0.6–1.1) | 0.7 (0.5–1.0) |

[b]Direct standardized for the age and sex groupings in the 2020 U.S. census.
[a]Estimates are weighted to the U.S. adult population.
[c]Model adjusted for sex and age; estimates are adjusted for age only.
[d]Aged 65 or older OR ≥ 1 comorbidity OR unvaccinated.

*Vulnerability to severe COVID, awareness and uptake of antivirals among those with SARS-CoV-2 infection.* Among the 22.1% with SARS-CoV-2 infection during the study period, 74.5% (95% CI 65.4–83.6%) had one or more vulnerability, 66.2% (95% CI 55.7–76.7%) had hybrid protection, and 29% (95% CI 19.6–38.6%) met eligibility criteria for antivirals (by virtue of being aged 65+ or having one or more comorbidities)[22] (Table 3). Of those who tested with a healthcare or testing provider, 55.9% (95% CI 44.9–67.0%) were not aware of the antiviral nirmatrelvir/ritonavir and 3.1% (95% CI 0.0–7.1%) reported that they tried to access it but could not.

Among those with a recent SARS-CoV-2 infection, 15.1% (95% CI 7.1–23.1%) reported receiving nirmatrelvir/ritonavir, with substantial variability by sociodemographic groups (Table 4). Reported nirmatrelvir/ritonavir use was higher among those with any medical vulnerability versus those without, among non-Hispanic white and Asian/Pacific Islander adults compared with Hispanic and non-Hispanic Black adults, those with health insurance versus those without, those under 65 years versus those older, those who were employed versus not, those who were college graduates versus those who were not, those with household income >25 K versus those with lower income.

**Routine surveillance of SARS-CoV-2 testing, cases, hospitalizations, and deaths**

*Testing and cases.* Figure 3a shows trends in SARS-CoV-2 tests administered by healthcare and testing providers with results reported to the NYC DOHMH among testers of all ages since the beginning of the pandemic through June 10, 2022. Throughout this entire period, 31.3 million PCR tests and 9.7 million rapid antigen tests were reported by providers and laboratories. PCR tests were the predominant test type reported (76%), comprising 83% of all positive tests. PCR tests remained predominant for the period since October 2020 when rapid antigen tests were more

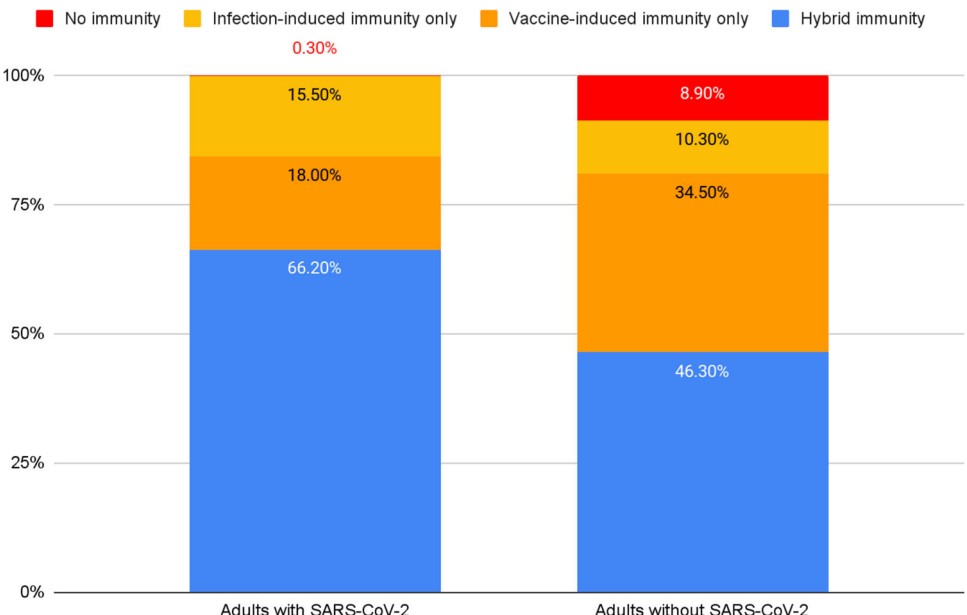

**Fig. 2 Hybrid immunity among adults with and without SARS-CoV-2 infection, NYC April–May 2022.** Estimates were obtained by combining self-reported information on uptake of SARS-CoV-2 vaccination with history of prior SARS-CoV-2 infection.

widely available at testing locations and comprised 73% of all tests and 80% of all positive tests. There were substantial declines in SARS-CoV-2 testing per 100,000 residents leading up to the study period (Fig. 3a). During the study period (April 23–8 May 2022), there were 671,377 tests reported, which gave rise to 51,218 diagnosed and reported cases (7.6% of tests). Of all tests during the study period, 505,242 (75%) were PCR tests, of which 33,066 (6.5%) were positive, and 166,135 (25%) were rapid antigen tests, of which 9076 (5.5%) were positive.

*Hospitalizations and deaths.* COVID-19 hospitalizations and deaths increased during the BA.2/BA.2.12.1 surge, but modestly in comparison to that during the BA.1 surge during December 2021–February 2022 (Fig. 3b). During 1 December–14 March, which brackets NYC's BA.1 surge, there were 33,683 hospitalizations and 4867 deaths. But during 15 March–20 June, which includes the BA.2/BA.2.12.1 surge to date, the cumulative number of hospitalizations and deaths were much lower (6983 hospitalizations and 651 deaths).

**Wastewater surveillance**. The trend in SARS-CoV-2 wastewater concentration from the 14 NYC WRRFs is shown in Fig. 4. The largest peak in SARS-CoV-2 concentration in wastewater is for the BA.1 surge in December 2021–January 2022, followed by a more modest peak during the BA.2/BA.2.12.1 surge in March–June 2022. The BA.2/BA.2.12.1 surge was similar to the peak of the Delta surge in the fall of 2021. The ratio of SARS-CoV-2 wastewater concentration to reported cases is shown in Supplementary Fig. 1. The mean peak viral concentration per capita for the BA.1 surge across all 14 WWRFs was 58,294,503 N1 copies per liter per day per capita on 26 December 2021, and the peak daily case count was 60,757 on 3 January 2022 (ratio of peak wastewater concentration: peak cases = 959.5). For the BA.2/BA.2.12.1 surge, these numbers were 10,743,571 N1 copies per liter per day per capita on 3 May 2022, and 5664 cases on 23 May 2022 (ratio of mean peak wastewater concentration: peak cases = 1896.8), which corresponds to a two-fold higher mean peak concentration of SARS-CoV-2 in wastewater per reported SARS-CoV-2 case than that during the BA.1 surge.

## Discussion
Our survey found a much higher prevalence of active SARS-CoV-2 infection during the BA.2/BA.2.12.1 surge in late April and early May 2022 than was detected by traditional case-based surveillance. We estimate that 22.1% of adult New Yorkers, ~1.5 million adults, had SARS-CoV-2 infection during the 2-week study period, when the more transmissible BA.2.12.1 subvariant made up an estimated 20% of all cases and was increasing rapidly[23]. The estimate of 1.5 million infections is about 29-fold higher than the 51,218 cases in the official NYC case counts[24] and suggests a potentially vast underestimate of the BA.2./BA.2.12.1 surge's magnitude, and a high incidence of reinfections and breakthrough infections. Importantly, while the numbers of COVID hospitalizations and deaths increased, they remained much lower than that during the recent BA.1 surge. Even though a high proportion of individuals vulnerable to a severe outcome were infected and did not use rapid antivirals, it appears that most also had some protection against a severe outcome through vaccination and boosters, on top of a history of prior infection. This high degree of hybrid immunity, coupled with high vaccine- and recently acquired infection-induced immunity via BA.1, if temporary, could partly explain why NYC did not experience a major increase in hospitalizations during the BA.2.12.1 surge.

We found substantial differences in age- and sex-adjusted SARS-CoV-2 prevalence by sociodemographic factors, including race/ethnicity, which could be reflective of a number of things, alone or in combination, including greater exposure to SARS-CoV-2 (i.e., in the home, workplace or other setting(s)), and differences in individual behaviors around masking and social distancing. Household characteristics (number of household members and children) and individual behaviors may be increasingly relevant as a determinant of infection risk during surges going forward, as many pandemic restrictions[25] had been recently dropped in NYC, leaving decisions about COVID precautions up to individual citizens.

We estimated a higher age- and sex-adjusted prevalence of SARS-CoV-2 infection among those who were boosted compared with those who were fully vaccinated but not boosted and those who were unvaccinated. Since vaccines and boosters provide limited protection against *infection* with omicron, these

**Table 3 Characteristics and antiviral awareness and uptake among NYC adults with SARS-CoV-2 infection, April–May 2022**

| | Total |
|---|---|
| | % (95% CI) |
| Total number of positives | 227 (100) |
| Hybrid immunity | |
| Vaccine- and infection-induced | 66.2 (55.7–76.7) |
| Vaccine-induced only | 18.0 (8.4–27.6) |
| Infection-induced only | 15.5 (8.2–22.7) |
| No immunity | 0.3 (0.0–0.8) |
| Vulnerability to severe COVID-19 | |
| Any vulnerability[a] | 74.5 (65.4–83.6) |
| Unvaccinated | 15.8 (8.5–23.0) |
| Prior history of COVID | 98.2 (94.6–100) |
| No prior history of COVID | 1.8 (0.0–5.4) |
| Comorbidity[a] | 57.8 (47.5–68.1) |
| Age 65+ | 12.7 (8.8–16.6) |
| Antiviral eligibility and awareness | |
| Eligible for antivirals[b] | 29.0 (19.6–38.4) |
| Awareness (n = 192) | |
| Unaware of antivirals | 55.9 (44.9–67.0) |
| Aware of antivirals | 44.1 (33.0–55.1) |
| Tried to access but could not | 3.1 (0.0–7.1) |

[a]Aged 65 or older OR >1 comorbidity OR unvaccinated; possible comorbidities included: cancer, diabetes, obesity, COPD or lung disease, liver disease, heart disease, high blood pressure, a recent organ transplant, or an immunodeficiency.
[b]Eligible: above 65 or with comorbidities, with reported symptoms and tested positive on at-home rapid or POC rapid or PCR test.

**Table 4 Nirmatrelvir/ritonavir uptake among NYC adults with SARS-CoV-2 infection, April–May 2022**

| | Total |
|---|---|
| | % (95% CI)[a] |
| Overall | 15.1 (7.1– 23.1) |
| Uptake by subgroup | |
| Vulnerable to severe COVID-19 | |
| Yes | 17.4 (7.5–27.3) |
| No | 6.9 (0.0–15.3) |
| Age | |
| 65+ | 2.1 (0.0–5.4) |
| <65 | 17.2 (7.8–26.5) |
| Age | |
| 55+ | 7.0 (0.13–13.9) |
| <55 | 18.7 (7.5–29.8) |
| Race/ethnicity | |
| NH White | 19.7 (7.9–31.4) |
| NH Black | 6.5 (0.0–20.0) |
| Hispanic | 11.5 (0.0–25.7) |
| Asian/Pacific Is. | 52.7 (0.0–100) |
| Health insurance | |
| Yes | 17.3 (7.9–26.7) |
| No | 3.8 (0.0–11.7) |
| Employed | |
| Yes | 22.6 (10.1–35.2) |
| No | 2.9 (0.0–7.2) |
| Education | |
| College graduate | 32.7 (16.5–48.8) |
| Not college graduate | 3.3 (0.0–7.1) |
| Household income | |
| >25 K | 17.8 (8.2–27.4) |
| ≤25 K | 3.3 (0.0–10.2) |

[a]Uptake only assessed among 192 positive respondents who were in contact with a healthcare or testing provider. Precision of estimates is low due to smaller sample size.

differences are likely due to differences in SARS-CoV-2 exposures and behaviors between the two groups. These findings have important implications for observational (test negative) vaccine effectiveness (VE) studies, which are confounded by differences in exposure/behavior, testing behavior, and history of prior SARS-CoV-2 infection between those vaccinated/boosted and unvaccinated. Given that the prevalence of prior SARS-CoV-2 infection was nearly 62% among unvaccinated persons, not taking into account the likely differential depletion of 'susceptibles' by vaccination status (i.e., due to SARS-CoV-2 infections) and differences in testing behaviors will bias (underestimate) VE against severe disease and death in test negative designs[26]. Survey data such as ours can be used to correct VE estimates for these biases in terms of addressing selection bias as it relates to testing and health-seeking behaviors (i.e., those who are motivated to be tested and vaccinated are more likely to access health services).

When we took into account both vaccination status and prior SARS-CoV-2 infection, we found that those with hybrid immunity had higher adjusted SARS-CoV-2 prevalence (29.2%) than those who had vaccine-induced protection only (12.9%) and a similar prevalence to those with infection-induced protection only (24.6%). This suggests that prior infection (more so than vaccination) is a strong marker for exposure risk during surges (e.g., workplace, household) and possibly reflects a lower perceived risk for infection/reinfection, severe disease/death, and onward transmission and/or less ability to avoid exposure[27]. The possible role of past and more recent SARS-CoV-2 infections in reducing adoption of personal risk mitigation measures during a surge needs to be further examined. Increasing first, second and subsequent vaccine doses among those with any history of previous infection is therefore a key strategy to help lower the population risk of severe COVID and death.

We performed a sensitivity analysis to examine whether the higher prevalence of SARS-CoV-2 among vaccinated and boosted and among those with hybrid immunity were driven by likelihood of test seeking rather than likelihood of being infected. We found that among testers only (n = 554), prevalence of infection was still higher among vaccinated and boosted compared to those fully vaccinated not boosted and those who were not vaccinated. Similarly, prevalence of infection was higher among those who had hybrid immunity than those who had vaccination-induced immunity only, infection-induced immunity only, and those with no protection. These findings suggest that differences in prevalence by vaccination status or hybrid protection are not due to differences in test seeking behaviors or access.

A substantial proportion (27.8%) of adults who are vulnerable to a severe SARS-CoV-2 outcome were estimated to have active SARS-CoV-2 during the BA.2/BA.2.12.1 surge, reinforcing the importance of vaccination and boosters in this group. Observational studies report that vaccine effectiveness against hospitalization with omicron (BA.1) is ~55% for two doses and 80% after a single booster, soon after dosing, and worsens substantially by 3 months[28,29]. While recent prior infection with BA.1 does not necessarily protect against re-infection with BA.2 or BA.2.12.1[30,31], it may confer enhanced protection against severe disease. The fraction of individuals who would likely be hospitalized with COVID-19 following recent prior omicron infection is unknown. However, in the pre-vaccine era in NYC, hospitalization would have been expected for 3.7% of re-infected individuals[32].

Our study suggests that awareness and uptake of nirmatrelvir/ritonavir (Paxlovid™) was low among adults with SARS-CoV-2 infection in our study. Nirmatrelvir/ritonavir trials were conducted among unvaccinated individuals at high risk for

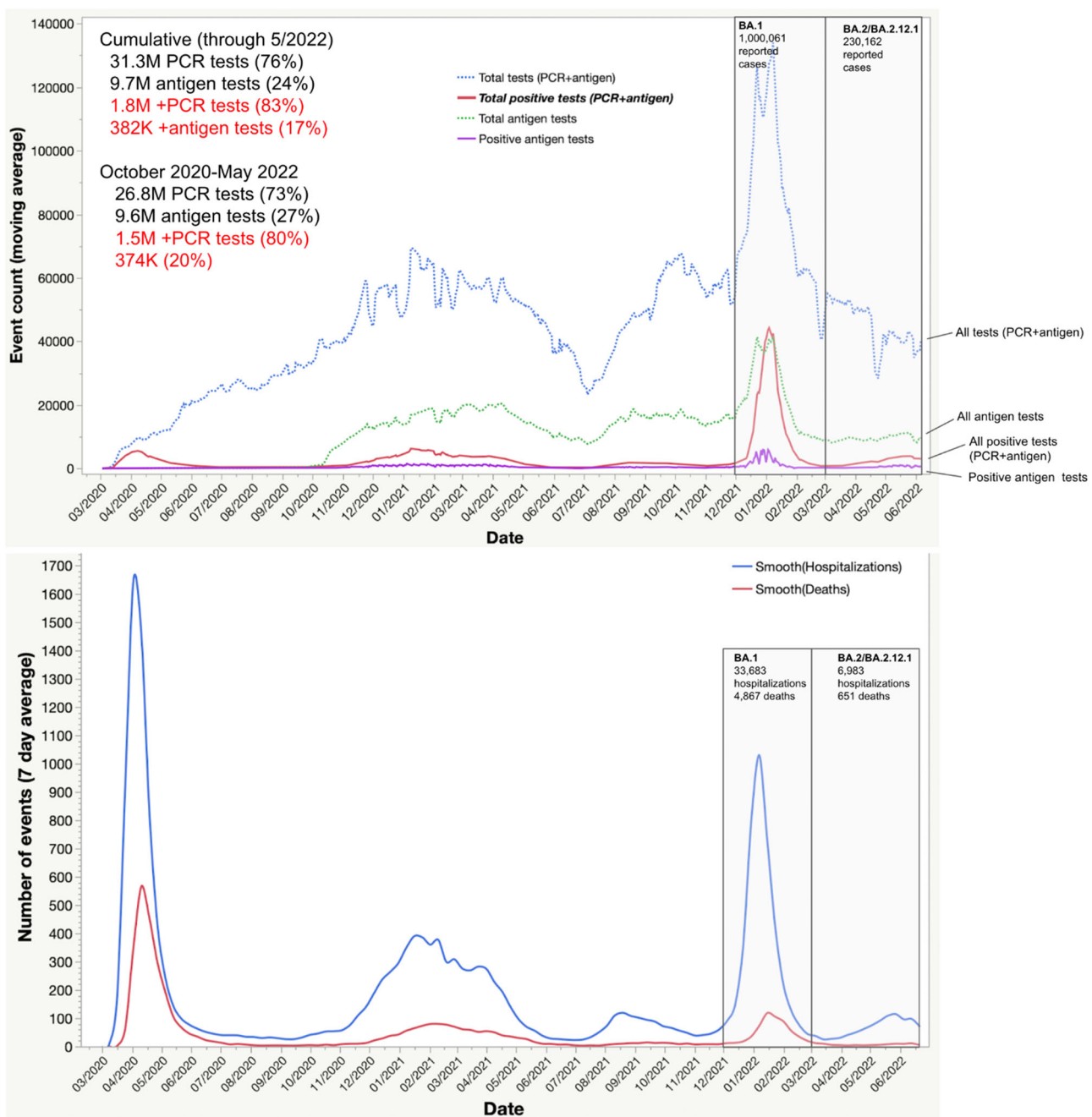

**Fig. 3 Total PCR tests, antigen tests, positive PCR tests, positive antigen tests, COVID-19 related hospitalizations and deaths, NYC.** Variant eras for NYC were approximated based on the timing of peaks and troughs in COVID-related hospitalizations in NYC as follows: 1 December 2021–1 March 2022 (BA.1) and 1 March 2022–6 June 2022 (BA.2/BA.2.12.1).

hospitalization and death, and were shown to reduce the likelihood of these outcomes by ~90%[3]. It is unclear how much added protection nirmatrelvir/ritonavir provides over and above that provided by vaccines/boosters. A study among vaccinated adults who received nirmatrelvir/ritonavir had reductions in ER visits, hospitalization, or death[33]. CDC currently recommends[34] antivirals for individuals susceptible to severe COVID-19, regardless of vaccination status.

Our sample size was small in analyses restricted to those with SARS-CoV-2 infection. However, stratified analyses of nirmatrelvir/ritonavir uptake by sociodemographic factors and biologic vulnerability suggested that there may be important inequities in antiviral access across a number of social determinants of health. A recent national CDC study had similar findings[35]. While

caution is warranted in the interpretation of our estimates of these inequities, as the confidence limits were wide, these are potentially important findings that warrant further investigation and monitoring, with policy and programmatic course corrections as needed. Inequitable uptake of antivirals among vulnerable individuals with COVID-19 will further exacerbate inequities in the burden of SARS-CoV-2 which has had disproportionate effects on racial/ethnic minorities and other groups[36–38]. It is essential to give attention to both need and equity in the design and implementation of large scale public health initiatives, and to avoid designs were such initiatives may create new inequities or exacerbate existing inequities.

The signal of SARS-CoV-2 concentration per capita in NYC wastewater surveillance data, which covers an estimated

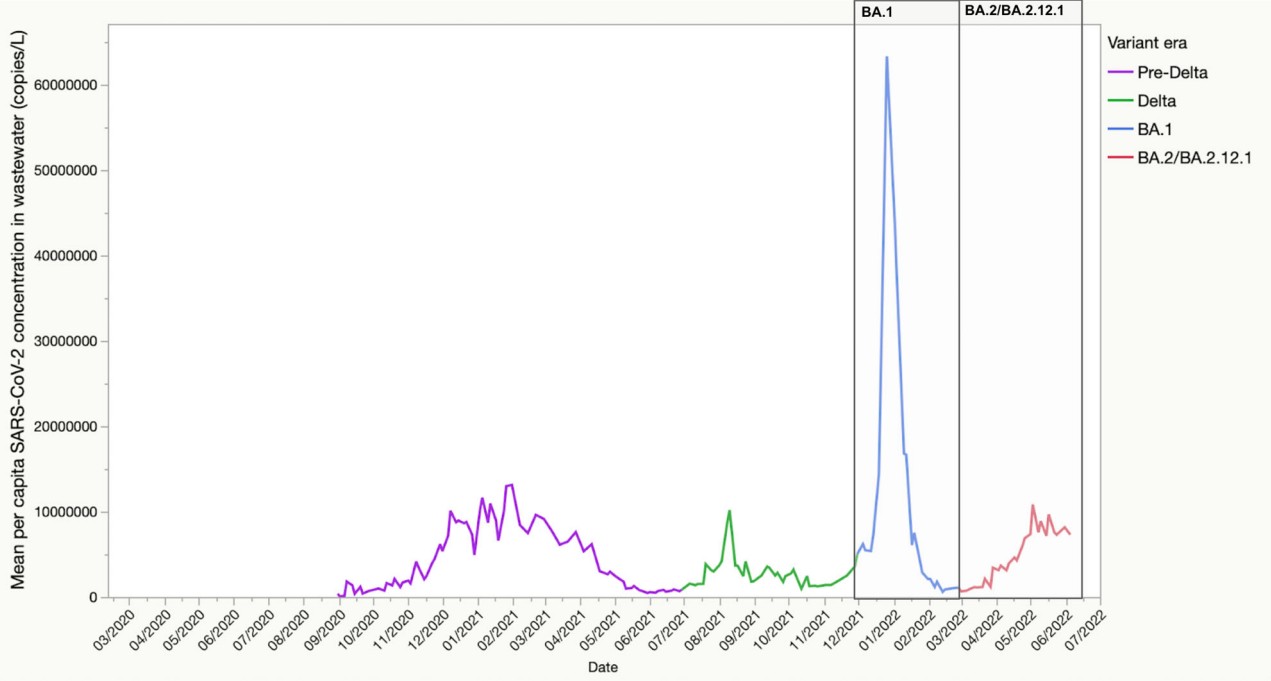

**Fig. 4 Mean per capita SARS-CoV-2 concentrations from 14 water resource recovery facilities (WRRFs) in NYC.** WRRFs in NYC cover wastewater of an estimated 8.2 million residents. WRRFs are sampled up to twice each week and per capita SARS-CoV-2 load (N1 copies per capita) is reported for each sample date. We plotted the mean per capita SARS-CoV-2 load by sample date across all 14 WRRFs. Variant eras for NYC were approximated based on the timing of peaks and troughs in COVID-related hospitalizations in NYC as follows: 1 December 2021–1 March 2022 (BA.1) and 1 March 2022–6 June 2022 (BA.2).

8.2 million New Yorkers, was modest in comparison to that during the BA.1. surge. When we accounted for the number of reported cases, the mean SARS-CoV-2 concentration per reported case in wastewater was twice as high during the BA.2/BA.2.12.1 surge vs. BA.1. This could reflect an increase in the mean peak viral load per case between the two subvariant surges (e.g., a result of the properties of the pathogen and/or host susceptibility), and/or a decline in the completeness of case reporting coupled with an increased reliance on at-home rapid testing. The latter seems a more plausible explanation, given the similarities of BA.1 and BA.2/BA.2.12.1. Rather, because of recent prior BA.1 infection among a wide swath of the NYC adult population, it may be that those re-infected with BA.2/BA.2.12.1 after a recent BA.1 infection had a lower mean peak viral load due to greater acquired T-cell mediated immunity from prior BA.1 infection[39–41]. This could effectively reduce the magnitude of peak SARS-CoV-2 concentration in wastewater, even for roughly the same number of individuals in the population with active SARS-CoV-2 infection.

Indeed, a very large pre-Omicron study from Qatar compared cycle threshold (Ct) values of over 380,000 individuals with active primary, reinfection, or breakthrough infection, adjusting for sex, age, reason for testing and calendar week of testing. Compared with unvaccinated people with primary infection, Ct values were 4.0 cycles higher in unvaccinated people who had reinfection, followed by vaccinated people with a breakthrough infection (1.3 cycles higher for BNT162b2 breakthrough infections and 3.2 cycles higher for those with mRNA-1273 breakthrough infections)[42]. A more recent study of those with BA.1 or BA.2 infection found that while those with BA.2 had adjusted cycle thresholds that were 3.5 cycles lower compared with BA.1, having a prior infection <90 days before a reinfection was associated with adjusted cycle thresholds that were 4.23 cycles higher[43]. These studies suggest that peak viral loads and/or length of shedding is

lowest in those with a reinfection. There are no equivalent studies for breakthrough infections among those who had both a prior infection and were vaccinated (hybrid immunity), but it is possible that cycle thresholds could be higher.

We found a disconnect between the number of tests and positive tests with a healthcare provider in our survey and those reported in official SARS-CoV-2 surveillance data. About 43% of adults in our survey reported having received a SARS-CoV-2 test with a health care or testing provider in the prior 2 weeks, which would correspond to ~2.9 million adults tested during 23 April–8 May 2022. However, according to NYC surveillance data, only ~670 K tests were performed during the study period[24], suggesting that testing itself (and possibly rapid testing specifically) may be severely underreported by laboratories and providers (by a factor of 4), overestimated by our survey, or a combination of both. Passive surveillance relies on institutions to voluntarily report data (often in batches using electronic systems). However, data quality, timeliness and completeness often cannot be guaranteed and can be variable. If tests, including positive tests, are underreported, this could be part of the reason for the larger than expected discrepancy between case counts and our estimate of SARS-CoV-2 infections. Unfortunately, our survey (Supplementary Note 1) did not distinguish between PCR and rapid tests with provider-based testing. However, in a recent electronic health record (EHR) based analysis of data from a large NY area urgent care provider, our team found that rapid antigen tests were more common than PCR tests, including among positives[44], which conflicts with official city-wide data on testing. To our knowledge, the completeness, representativeness, timeliness, and acceptability of passive SARS-CoV-2 reporting of such a high volume of cases and test results by providers and laboratories, including during surges, has not been systematically assessed in NYC or elsewhere around the U.S., making it important to investigate this discrepancy. A recent analysis of Omicron infections over a five day

period (13–17 December, 2021) in England from the REACT-1 study estimated that the community SARS-CoV-2 prevalence (based on PCR swab positivity in study participants) was about 600,000, and this compared with 206,295 confirmed and probable Omicron cases diagnosed and reported through routine case surveillance from 1 to 21 December, suggesting substantial under ascertainment of cases in a setting where there are functional national systems for capturing the results of provider-based testing[45]. In general, passive surveillance of infectious diseases in the U.S. that relies on reporting by health care providers has low completeness, especially when there is an administrative burden to complete forms or enter data[46]. However, the completeness of laboratory reporting (i.e., of tests and related results) that leverages laboratory information systems can be higher. Completeness of both types of reporting is challenging to assess and improve. However, during public health emergencies, completeness can be enhanced via increased awareness and reminders to providers and laboratories around reporting obligations. Large surges in cases could create added administrative burden and demand on providers and laboratories, potentially reducing the proportion of tests and cases reported or reducing data quality/completeness. This may be particularly true for point of care tests during intense surges. Of note, effective April 4, 2022, HHS, CDC and New York State announced changes to laboratory reporting requirements that no longer required reporting of negative or inconclusive rapid antigen tests[47].

Our survey estimates of SARS-CoV-2 prevalence and provider testing are subject to selection and information biases. In terms of selection bias, our survey estimates may be biased due to non-response if those who responded differ from non-respondents. We could not correct for this bias; however, characteristics of survey respondents did not differ substantially from that of the adult NYC population (Supplementary Results). In addition, our SARS-CoV-2 prevalence estimates could be inflated if those who both tested for SARS-CoV-2 infection and tested positive were more likely to participate in the survey than those who did not. While potential survey participants were not aware of the survey content before deciding to participate, it may be that those who were positive were more likely to complete the survey. Our survey response rate was low (1.2%) given that all sampling was done through phone-based random digit dialing (see Supplementary Results). While phone-based surveys have increasingly been subject to low response rates over time[48], studies have shown that response rates are a poor indicator of non-response bias and data quality[49]. In terms of information bias, it is also possible that participants inadvertently recalled and reported positive tests that were beyond the 14-day study period (recall bias). Lastly, some people test multiple times with providers after their initial positive test[42], and subsequently, many can expect positive PCR and antigen test results for 10 or more days[46,47]. This could have caused some people who were diagnosed prior to the study period to have positive tests during the study period which could have inflated our prevalence estimates relative to official case counts. But from an epidemiologic standpoint, these individuals with positive antigen test results should be reflected in prevalence estimates as they are actively infected that could result in hospitalization or onward spread.

While non-response bias could affect our survey prevalence estimates, we believe population-representative surveys like ours have an important value in that they contribute to the understanding of where the burden of infection stands during a surge across subpopulations in a current landscape where there is increasingly limited data that are not subject to testing bias[7]. Even if the true prevalence estimate of SARS-CoV2 is closer to the lower bound, the burden is still higher than what is reflected in standard surveillance. Cross-sectional prevalence surveys also provide an important snapshot of which vulnerable groups are least likely to test and those who have the highest burden of infection, since they can collect more data than is typically gathered as part of routine diagnostic testing activities.

Our study had other limitations, including a limited sample size, especially in subgroups of those with evidence of active SARS-CoV-2 infection. For those with prior COVID, we did not capture information on timing of prior infections, which may overestimate the degree of current hybrid immunity, though a substantial proportion of NYC adults were infected during the recent BA.1 surge[10,24]. Our case definition would likely capture some of the estimated 20–30% of individuals whose SARS-CoV-2 infection may remain asymptomatic throughout their infection[50,51], as well as those who were symptomatic but were not aware of a close contact. Finally, our survey did not include children or those whose primary language was not English or Spanish.

Survey-based approaches to measure population-representative prevalence estimates could be vastly enhanced if these surveys are strategically and routinely deployed and combined with biomarker indicators[7]. Surveys that have incorporated PCR testing in prevalence estimation could validate the prevalence estimates based on self-report and asymptomatic cases that would otherwise be missed[52]. Routinely deployed population-representative cross-sectional surveys are also better equipped to detect surges[45] and to triangulate with wastewater-based surveillance so that trends may also be compared. If combined with serological testing, routinely deployed seroprevalence surveys can provide time-series trends[53] that can elucidate levels of population immunity due to prior infections and vaccinations[54]. However, it is also possible that the addition of biomarkers and related procedures could reduce participation and timeliness. A hybrid approach that includes biomarkers as a validation procedure could be used to correct self-reported data in the same survey or in similar surveys that do not use biomarkers.

Strengths of our study include the representative and probability-based design of the survey, the ability for the survey to reflect outcomes among those who do not access the healthcare system, and triangulation with other important data sources such as routine passive surveillance for SARS-CoV-2 cases, hospitalizations, and deaths, as well as wastewater surveillance. Other strengths include the study's timing at the start of the BA.2/BA.2.12.1 surge, and measurement of several important factors that are not currently available through routine surveillance, including estimation of: 1) outcomes among those who do not test at all or test exclusively at home during a surge; 2) prevalence among individuals vulnerable to COVID-19 3) hybrid immunity; and 4) awareness/uptake of nirmatrelvir/ritonavir.

## Conclusions

Our study characterized and quantified the extent to which the magnitude of NYC's BA.2/BA.2.12.1 surge was underestimated by official case counts due to a combination of exclusive at-home testing, not testing at all, incomplete provider/laboratory reporting. This is underestimation of burden likely also occurred in other U.S. jurisdictions, and by extension, the national SARS-CoV-2 surveillance system. Even though many individuals vulnerable to a severe outcome were infected and did not use rapid antivirals, most also had a high degree of protection against a severe outcome through vaccination and boosters, overlaid by a recent history of prior BA.1 infection.

At the outset of SARS-CoV-2 variant surges, given the uncertainty of how they will impact severe outcomes among those who remain susceptible, a shift in approach to public health surveillance for SARS-CoV-2 is needed. More routine and timely

quantification of SARS-CoV-2 infection burden can allow for the monitoring of the ratio of infections to hospitalizations and deaths, and thus characterize the severity of surges, including those due to novel variants. Our findings demonstrate the utility of population-representative surveys as an important surveillance tool to go alongside, and triangulate with, passive case reporting, genomic surveillance, and wastewater surveillance at uncertain and evolving stages of the U.S. pandemic.

## Data availability

Deidentified survey data are available on request. SARS-CoV-2 routine testing and case surveillance data and SARS-CoV-2 wastewater surveillance data are publicly available at the following links: 1. Daily data on SARS-CoV-2 cases, hospitalizations, and deaths in NYC (including source data for Fig. 3) are available at: https://github.com/nychealth/coronavirus-data/tree/master/trends#data-by-daycsv 2. Data on SARS-CoV-2 concentrations measured in NYC Wastewater (including source data for Fig. 4) are available at. NYC Open Data https://data.cityofnewyork.us/Health/SARS-CoV-2-concentrations-measured-in-NYC-Wastewat/f7dc-2q9f/data

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

## Acknowledgements

The authors wish to acknowledge the survey participants and Consensus Strategies for completing survey sampling and data collection. Funding for this project was provided by the CUNY Institute for Implementation Science in Population Health (cunyisph.org).

## Competing interests

D.N., M.R., and S.K. report support from a SARS-CoV-2 research grant from Pfizer to their institution. D.N. reports consulting fees from Abbvie and Gilead. The remaining authors declare no competing interests.
