## [Peer Review File · Communications Medicine]

Reviewers' comments:

Reviewer #2 (Remarks to the Author):

1. Brief summary of the manuscript

This manuscript describes the results of a rapid representative household survey performed in May 2022 in NY, with regards to prevalence of SARS-CoV-2 infection (predominantly self-reported), stratified by many relevant demographic and prior immune exposure characteristics.

2. Overall impression of the work

The manuscript describes a unique approach for quickly assessing burden of infections. This is especially valuable given that routine surveillance may no longer be adequate for picking up the breadth of infections given increased home test use and potential limitations in reporting. Strengths include the triangulation of multiple data sources and incorporation of complementary subject matter such as antiviral awareness and use. Limitations are the inability to control for a number of biases that may be contributing to a number of unusual findings. While limitations are discussed, this section could be strengthened.

3. Specific comments, with recommendations for addressing each comment

General – while the response rate was added to the response to prior reviewers, it was not added to the actual manuscript. Please include this, as it's helpful for understanding the potential limitations.

In regards to the line (pg 5-6): "Age and sex-adjusted SARS-CoV-2 prevalence increased in dose response fashion with the number of household members, and households with children <18 years had substantially higher prevalence than those without (31.5% 95%CI 12.9%-43.0% vs 17.8%, 95%CI 13.6%-22.8%) (Figure 1)" please clarify if the proportions presented in the statement are the prevalence among "households" or "members of households with children < 18years." If so, please update the statement to say this. Also, please note that the proportions listed in Figure 1, which appear to be at the adult participant-level, do not match the proportions included in this statement.

In the following statement (pg 6), "Those who said they tested positive for SARS-CoV-2 once (36.9%, 95%CI 27.6%-47.3%) or more than once (37.5%, 95%CI 27.2%-49.1%) had much higher age and sex adjusted prevalence than those who said they never tested positive before (11.5%, 95% CI 6.7%-18.9%) or who thought they had COVID before but never tested positive (13.0%, 95%CI 8.2%-20.0%)," please clarify if this entire statement is referring to people who had reported testing positive prior to the current episode. If so, please change to "tested positive for SARS-CoV-2 once or more than once before the current episode."

Given the marked difference in testing patterns by demographic and prior immune history factors, it is difficult to know how many of the differences in prevalence of SARS-CoV-2 infection listed on page 6 are being driven by likelihood of seeking testing as opposed to likelihood of being infected. The former would explain the higher rate of infection among those who are boosted and/or with hybrid immunity than those without in this study and the

very low rate of infection amongst those with “no immunity.” Although it’s impossible to fully tease this out with the data available here (without testing or serology data) – given only half of participants were tested and that testing did differ substantially between subgroups, it would be helpful to know if many of the key findings regarding comparative risk of infection persisted or disappeared if a sensitivity analysis were performed restricted to those who had tested. Would also consider adding this difference in testing behaviors to the explanatory considerations on page 10 and 11.

I have concerns with the paragraph that follows and appears to argue that this rapid survey approach can address biases related to differences in exposure and/or testing behaviors. As noted above, if a certain proportion of the population never tests for COVID-19 then they are less likely to identify their infection and report themselves positive on a survey. This seems highly possible/probable in a home setting such given that many upper respiratory tract infections come and go without diagnosis. While studies with a TND design, cohorts with serial testing, and seroprevalence studies all have their own limitations they do attempt to address this testing issue in different ways, whereas the approach used here doesn’t currently include any way of addressing this concern. Would consider removing the statement “Survey data such as ours can be used to correct VE estimates for these biases” or clarifying further.

Pg 12 - Since this manuscript was developed, there have now been a few articles showing a benefit of nirmatrelvir/ritonavir among vaccinated individuals (Effectiveness of nirmatrelvir-ritonavir against hospital admission: a matched cohort study in a large US healthcare system | medRxiv; Oral Nirmatrelvir and Ritonavir in Nonhospitalized Vaccinated Patients With Coronavirus Disease 2019 (COVID-19) | Clinical Infectious Diseases | Oxford Academic (oup.com))

Regarding the statement on page 16 “But from an epidemiologic standpoint, these individuals should be reflected in prevalence estimates as they are actively infected” it is not clear what is meant by “actively infected,” but in most studies people who are immunocompetent no longer shed viable/culturable virus beyond 10-12 days, whereas the PCR test can remain positive for weeks after. (Comparison of Home Antigen Testing With RT-PCR and Viral Culture During the Course of SARS-CoV-2 Infection | Infectious Diseases | JAMA Internal Medicine | JAMA Network)

Reviewer #3 (Remarks to the Author):

The work estimates crude prevalence rates of SARS-CoV-2 infections in the last two weeks using a survey-based approach in NYC. Based on integration of questions on risk factors, the authors further identify potential contributors to increased risk of infection including increasing household size and children under-18 (likely confounded with household size). Estimated rates are then qualitatively compared to other data sets, including reported case data, hospitalizations, deaths, and SARS-CoV-2 RNA concentrations in wastewater. The authors estimate that there is a 30-fold higher estimate of infections in the prior two weeks than is reported in the official case count, and that the magnitude of the surge in BA.2 infections was significantly underestimated by alternative monitoring methods. The authors

then extend their findings to suggest that immunity helped to prevent a surge in hospitalizations and deaths. The methodology provides insights into the extent to which survey-based data on infections may be useful for estimating crude-prevalence rates.

1. There is clear bias in survey-based metrics (non-response, social desirability), particularly given the low response rate. Although the authors acknowledge this, the manuscript would benefit from a more nuanced discussion of their impacts on the study conclusions. If the survey-based approach shows dramatically higher incidence, should the survey results be taken as the standard metric to which all other metrics are insufficient? I would argue that no, the true (unobservable) prevalence rate is likely between the upper and lower bounds. If so, what additional value does phone-based surveys provide?

2. A discussion of the proposed methods relative to other methods that complemented cross-sectional surveys with objective indicators (PCR-testing, serology) (see Stringhini et al. 2020 and 2021 in Geneva; Layton et al. 2022 in Oregon) is warranted, particularly on the relative reliability of the resulting estimates.

3. The qualitative comparisons to other metrics is limited by the single point prevalence estimate available in the cross-sectional survey relative to the historical time series data of other metrics. All of the data sources are limited, as the authors note, but the time series data benefits from temporal (dynamic) comparisons. Given the relative ease and low-cost of phone-based surveys, such an approach could easily be extended to gather time series data that would be more comparable. See minor comments on limitations of the qualitative comparisons to wastewater.

Minor comments:

“intervals between surges”, which is this distinct from “population levels of immune protection”?

“22.1% of respondents”, having a sample size here would be useful.

“We plotted the mean per capita SARS-CoV-2 load (n1 copies per day per population)”.

Capita and population are reported twice in this description. Load is typically reported as copies per day. This may be an error, or it may be a miscalculation. The visualization of mean per capita SARS-CoV-2 load assumes temporal alignment of disease dynamics with normal distributions, which is not likely accurate because there are likely spatially-resolved heterogeneity in timing of peaks and troughs. Have the authors considered visualizing a summed loading? The dynamics are likely similar, but this would be more mechanistic (“total load”).

Figure 1, why not include error bars to highlight variation and uncertainty, in line with recommendation for data presentation guidelines (clearly defined error bars).

Figure 4; Define Variant Era (50% or more samples based on clinical sample sequencing?) in the Figure caption.

Figure S1 has a strange formatting issue at 1000. Was this image overlaid with a line or otherwise formatted strangely? What is this ratio describing? The data in Figure 4 or the loads? A more clear description of the aggregation of data from the 14 WWTPs would be beneficial in this graphic.

“as many pandemic restrictions...” suggest providing citations for this.

“Despite our estimate...” Suggest the authors revise this section. The authors did not conduct a survey during BA.1, and so can not draw conclusions about the relative number of people infected during BA.1 compared to their estimated 1.5M during 2 weeks of BA.2. The current study does not detect a “surge”, it only estimates a crude prevalence. Although they find 1.5 million people were infected in the last two weeks, the relatively high wastewater loads

observed during this period (Figure 4) could explain this, but in the absence of a time series of survey-based methods, it is impossible to observe differences in survey to wastewater concentration ratios.

"If this is true, ", it appears from the figures that the wastewater signal performed at least as well in observing the surge as other time series indicators (reported cases, hospitalizations, and deaths). See prior comment on the inability of a cross-sectional survey to detect a surge. "Indeed, a very large... " What are the clinical samples discussed here? Although the authors make a strong and important point about the potential of prior infection or vaccination to attenuate the ratio between true (unobservable) infections and wastewater concentrations, as written the point of this section is not so clear.

"in a recent EHR", define HER.

"Our survey estimates of... "Non-response bias discussed here may play a role, acquiescence or social desirability biases may also artificially inflate estimates. See major comments.

"From an epidemiological standpoint, " Although it is true these people could be included as active shedders in estimates within the scope of infectious disease modeling, these authors otherwise do not include (or discuss) these people within the scope of other metrics used. In comparing authors results to other data sets, this should be held consistent or recognized as a clear limitation of the study's ability to compare.

"was underestimated by" it is generally well-accepted (and not novel) that official case counts represent a portion of true (unobserved cases). For wastewater, the discussion highlighting the insufficiencies is flawed (see prior comment), and the authors do not provide a basis for estimating incidence based on wastewater concentrations. Suggest the authors revisit this statement.

"Limiting the surge's impact" the authors attribute lack of hospitalizations and deaths to prior infection, but this does not follow directly on the research.

Wednesday, January 18, 2023

Dear Dr. Abbot,

We have revised our manuscript in response to the reviewer comments. These are detailed point by point below.

Sincerely,

Denis Nash, PhD, MPH
Distinguished Professor of Epidemiology, CUNY School of Public Health
Executive Director, CUNY Institute for Implementation Science in Population Health
Denis.Nash@sph.cuny.edu; Twitter: @epi_dude; 347-331-6554 (m)

Reviewers' comments:

Reviewer #2 (Remarks to the Author):

1. Brief summary of the manuscript

This manuscript describes the results of a rapid representative household survey performed in May 2022 in NY, with regards to prevalence of SARS-CoV-2 infection (predominantly self-reported), stratified by many relevant demographic and prior immune exposure characteristics.

Authors' response: Thank you for your review and feedback.

2. Overall impression of the work

The manuscript describes a unique approach for quickly assessing burden of infections. This is especially valuable given that routine surveillance may no longer be adequate for picking up the breadth of infections given increased home test use and potential limitations in reporting. Strengths include the triangulation of multiple data sources and incorporation of complementary subject matter such as antiviral awareness and use. Limitations are the inability to control for a number of biases that may be contributing to a number of unusual findings. While limitations are discussed, this section could be strengthened.

Authors' response: Thank you. We have strengthened the Discussion section where we discuss the limitations of the study, and describe them in response to specific comments below

3. Specific comments, with recommendations for addressing each comment

General – while the response rate was added to the response to prior reviewers, it was not added to the actual manuscript. Please include this, as it's helpful for understanding the potential limitations.

Authors' response: We have included the response rate in the discussion section as a limitation and have strengthened the section as suggested in comment above, as follows:

Discussion, Page 20:

Our survey estimates of SARS-CoV-2 prevalence and provider testing are subject to selection and information biases. In terms of selection bias, our survey estimates may be biased due to non-response if those who responded differ from non-respondents. We could not correct for this bias; however, characteristics of survey respondents did not differ substantially from that of the adult NYC population (Appendix 1). In addition, our estimates could be inflated if those who both tested for SARS-CoV-2 infection and tested positive were more likely to participate in the survey than those who did not. While potential survey participants were not aware of the survey content before deciding to participate, it may be that those who were positive were more likely to complete the survey. . Our survey response rate was low (1.2%) given that all sampling was done through phone-based random digit dialing (see Appendix 1). While phone-based surveys have increasingly been subject to low response rates over time⁴⁶, studies have shown that rates are a poor indicator of non-response bias and data quality⁴⁷. In terms of information bias, it is also possible that participants inadvertently recalled and reported positive tests that were beyond the 14-day study period (recall bias). Lastly, some people test multiple times with providers after their initial positive test⁴⁰, and subsequently, many can expect positive PCR and antigen test results for 10 or more days.^{44,45} This could have caused some people who were diagnosed prior to the study period to have positive tests during the study period which could have inflated our prevalence estimates relative to official case counts. But from an epidemiologic standpoint, these individuals with positive antigen test results should be reflected in prevalence estimates as they are actively infected that could result in onward spread.

In regards to the line (pg 5-6): “Age and sex-adjusted SARS-CoV-2 prevalence increased in dose response fashion with the number of household members, and households with children <18 years had substantially higher prevalence than those without (31.5% 95%CI 12.9%-43.0% vs 17.8%, 95%CI 13.6%-22.8%) (Figure 1)” please clarify if the proportions presented in the statement are the prevalence among “households” or “members of households with children < 18 years.” If so, please update the statement to say this. Also, please note that the proportions listed in Figure 1, which appear to be at the adult participant-level, do not match the proportions included in this statement.

Authors' response: Thank you for your comment. We have clarified in the results the proportions and 95% confidence intervals to reflect both the number of people in households and households with children < 18 years. We also correct the proportions in Figure 1 as they reflect the crude values and not the age- and sex-adjusted proportions.

Results, Page 10:

Age and sex-adjusted SARS-CoV-2 prevalence among survey respondents increased in dose response fashion with their number of household members (15.0% 95% CI 9.8%-22.6%) vs 21.1% 95% CI 15.9% - 27.5% vs 25.7% 95% CI 17.9% - 35.3%) , and respondents in households with children <18 years had substantially higher prevalence than those in households without children (31.5% 95%CI 12.9%-43.0% vs 17.8%, 95%CI 13.6%-22.8%) (Figure 1).

Figure 1. Age- and sex-adjusted SARS-CoV-2 prevalence estimates among NYC adults by household size and presence of children in the household, April-May, 2022

In the following statement (pg 6), “Those who said they tested positive for SARS-CoV-2 once (36.9%, 95%CI 27.6%-47.3%) or more than once (37.5%, 95%CI 27.2%-49.1%) had much higher age and sex adjusted prevalence than those who said they never tested positive before (11.5%, 95% CI 6.7%-18.9%) or who thought they had COVID before but never tested positive (13.0%, 95%CI 8.2%-20.0%),” please clarify if this entire statement is referring to people who had reported testing positive prior to the current episode. If so, please change to “tested positive for SARS-CoV-2 once or more than once before the current episode.”

Authors’ response: Thank you for your comment. We have clarified this as suggested.

Results, Page 11: Individuals who were fully vaccinated with a booster had higher age and sex-adjusted SARS-CoV-2 prevalence (25.2%, 95%CI 20.0%-31.3%) than those who were fully vaccinated but not boosted (11.8%, 95%CI 5.7%-23.0%) and those who were unvaccinated (15.3%, 95%CI 9.6%-23.5%). Those who said they tested positive for SARS-CoV-2 once before the current episode (36.9%, 95%CI 27.6%-47.3%) or more than once (37.5%, 95%CI 27.2%-49.1%) had much higher age and sex-adjusted prevalence than those who said they never tested positive before (11.5%, 95% CI 6.7%-18.9%) or who thought they had COVID before but never tested positive (13.0%, 95%CI 8.2%-20.0%).

Given the marked difference in testing patterns by demographic and prior immune history factors, it is difficult to know how many of the differences in prevalence of SARS-CoV-2 infection listed on page 6 are being driven by likelihood of seeking testing as opposed to likelihood of being infected. The former would explain the higher rate of infection among those who are boosted and/or with hybrid immunity than those without in this study and the very low rate of infection amongst those with “no immunity.” Although it’s impossible to fully tease this out with the data available here (without testing or serology data) – given only half of participants were tested and that testing did differ substantially between subgroups, it would be helpful to know if many of the key findings regarding comparative risk of infection persisted or disappeared if a sensitivity analysis were performed restricted to those who had tested. Would also consider adding this difference in testing behaviors to the explanatory considerations on page 10 and 11.

Authors’ response: Thank you for the suggestion. We have updated the table for prevalence by characteristics among testers (n=554). Differences in associations were observed across the following respondent characteristics:

1. Age distribution - among testers, higher prevalence among 25-34 and 45-54 year olds
2. Borough - among testers, highest in Brooklyn than the Bronx
3. Vaccination status - among testers, prevalence ratios among boosted and not-vaccinated compared to fully vaccinated but not boosted were attenuated than the estimate from the main analysis.
4. Immunity - among testers, prevalence highest among those with hybrid immunity, followed by vaccine-induced only then by infection-induced only.

We have updated the discussion to mention this sensitivity analysis, and that it suggests that differences in prevalence by vaccination status or hybrid protection are not due to differences in test seeking behaviors.

Discussion, Page 17: We performed a sensitivity analysis to examine whether the higher prevalence of SARS-CoV-2 among vaccinated and boosted and among those with hybrid immunity were driven by likelihood of test seeking rather than likelihood of being infected. We found that among testers only, prevalence of infection was higher among vaccinated and boosted compared to those fully vaccinated not boosted and those who were not vaccinated. Similarly, prevalence of infection was higher among those who have hybrid immunity than those who had vaccination mediated only, infection-mediated only, and those with no protection. These findings suggest that differences in prevalence by vaccination status or hybrid protection are not due to differences in test seeking behaviors.

(cont'd)

Table 1. Characteristics among **testers** for survey respondents by point prevalence of SARS-CoV-2, NYC April-May 2022

	Total	Crude prevalence of SARS-CoV-2 infection†	Age and sex direct-standardized prevalence of SARS-CoV-2 infection**	Crude prevalence ratio (PR)	Adjusted prevalence ratio (aPR)***
	Weighted N (%)	% (95% CI)	% (95% CI) (Weighted N = 991)	PR (95% CI) (Weighted N = 1,030)	aPR (95% CI) (Weighted N = 1,030)
Total	554 (100)	40.6 (33.9 – 47.3)			
Age					
18-24	74 (13.3)	41.9 (21.3 – 62.5)	38.3 (22.3 – 57.2)	0.9 (0.6 – 1.3)	0.9 (0.7 – 1.3)
25-34	108 (19.6)	46.2 (29.1 – 63.1)	44.2 (28.7 – 60.9)	-ref-	-ref-
35-44	80 (14.5)	43.4 (22.5 – 64.3)	43.9 (26.2 – 63.3)	0.9 (0.7 – 1.3)	0.9 (0.6 – 1.2)
45-54	101 (18.3)	43.6 (25.6 – 60.7)	44.5 (29.6 – 60.4)	0.9 (0.7 – 1.3)	0.9 (0.7 – 1.3)
55-64	93 (16.7)	39.0 (25.9 – 52.2)	38.0 (26.6 – 50.9)	0.8 (0.6 – 1.2)	0.8 (0.6 – 1.2)
65+	97 (17.5)	29.4 (22.1 – 36.6)	27.0 (20.7 – 34.5)	0.6 (0.4 – 0.9)	0.6 (0.4 – 0.9)
Gender					
Male	255 (46.0)	43.1 (32.6 – 53.7)	42.0 (32.6 – 52.1)	1.2 (0.9 – 1.4)	1.1 (0.9 – 1.4)
Female	281 (50.8)	36.8 (28.1 – 45.5)	37.0 (28.9 – 46.0)	-ref-	-ref-
Non-binary****	18 (3.2)	64.2 (36.8 – 91.8)	54.7 (41.9 – 66.9)	1.7 (1.2 – 2.6)	1.8 (1.2 – 2.7)
Race/Ethnicity					
Black NH	89 (16.0)	27.1 (13.5 – 40.7)	26.5 (16.4 – 38.9)	0.6 (0.4 – 0.9)	0.7 (0.5 – 1.0)
White NH	229 (41.4)	43.9 (34.6 – 53.1)	42.7 (34.4 – 51.4)	-ref-	-ref-
Hispanic	158 (28.6)	54.7 (42.1 – 67.3)	49.2 (39.2 – 59.3)	1.2 (1.0 – 1.5)	1.3 (1.1 – 1.6)
Asian/Pacific Islander	52 (9.4)	11.3 (0.0 – 24.8)	26.2 (16.9 – 38.3)	0.3 (0.1 – 0.6)	0.4 (0.2 – 0.7)
Other	25 (4.6)	30.1 (0.0 -60.7)	18.6 (7.6 – 38.4)	0.7 (0.4 – 1.3)	0.8 (0.4 – 1.4)
Years of education					
Some HS and below	93 (16.9)	50.6 (32.4 - 68.8)	46.2 (34.9 - 57.7)	1.6 (1.2 – 2.2)	1.8 (1.3 – 2.5)
HS Grad	157 (28.3)	32.2 (19.0 – 45.4)	30.2 (18.2 – 45.6)	-ref-	-ref-
Some college	102 (18.4)	33.0 (20.4 – 45.7)	35.1 (25.1 – 46.5)	1.0 (1.1 – 1.9)	1.1 (1.0 – 1.7)
College grad and above	201 (36.4)	46.3 (36.2 – 56.4)	46.0 (36.9 – 55.4)	1.4 (1.1 – 1.9)	1.3 (1.0 – 1.7)
Household size					
1	161 (29.1)	35.2 (23.2 – 47.2)	30.7 (20.6 – 43.1)	-ref-	-ref-
2-3	223 (40.4)	37.4 (27.9 – 46.8)	37.0 (28.2 – 46.7)	0.7 (0.6 – 0.9)	1.0 (0.8 – 1.3)
4+	169 (30.5)	50.0 (36.6 – 63.3)	42.8 (32.6 – 53.8)	1.4 (1.1 – 1.8)	1.3 (1.0 – 1.7)

(cont'd)

Any children <18 years					
Y	179 (32.3)	49.8 (36.5 – 63.1)	47.6 (35.0 – 60.6)	1.4 (1.1 – 1.7)	1.3 (1.1 – 1.6)
N	375 (67.7)	36.2 (28.7 – 43.7)	35.5 (27.9 – 43.9)	-ref-	-ref-
Household income					
Below 25K	135 (24.4)	30.5 (17.4 – 42.6)	21.4 (12.7 – 33.6)	0.5 (0.4 – 0.7)	0.6 (0.4 – 0.8)
25,000 - 65,000	149 (26.9)	46.3 (32.5 – 60.1)	47.4 (33.9 – 61.2)	0.8 (0.6 – 1.0)	0.8 (0.6 – 1.1)
65,000 - 150,000	124 (22.4)	57.9 (45.1 – 70.8)	61.1 (50.4 – 70.8)	-ref-	-ref-
Above 150,000	37 (6.7)	37.6 (15.6 – 59.7)	39.0 (23.9 – 56.5)	0.7 (0.4 – 1.0)	0.6 (0.4 – 1.0)
Prefer not to answer	109 (19.7)	26.6 (14.5 – 38.7)	25.8 (15.1 – 40.5)	0.5 (0.3 – 0.6)	0.5 (0.3 – 0.7)
Employed					
Yes	253 (45.8)	54.7 (44.9 – 64.5)	54.4 (45.8 – 62.8)	-ref-	-ref-
No/DK	300 (54.2)	28.7 (20.9 – 36.4)	25.5 (18.2 – 34.6)	0.5 (0.4 – 0.6)	0.5 (0.4 – 0.7)
Nativity					
Born in U.S.	398 (71.8)	47.5 (39.4 – 55.5)	46.4 (39.0 – 54.2)	-ref-	-ref-
Born outside U.S.	156 (28.2)	23.1 (13.7 – 32.4)	20.9 (12.7 – 32.5)	0.5 (0.4 – 0.7)	0.5 (0.4 – 0.7)
Borough					
Bronx	92 (16.7)	40.0 (23.0 – 56.9)	42.5 (30.8 – 55.1)	1.0 (0.7 – 1.4)	1.1 (0.8 – 1.5)
Brooklyn	169 (30.5)	50.5 (38.3 – 62.6)	48.8 (39.2 – 58.5)	1.3 (1.0 – 1.7)	1.2 (0.9 – 1.6)
Manhattan	114 (20.7)	40.0 (25.3 – 54.1)	39.2 (25.9 – 54.3)	-ref-	-ref-
Queens	148 (26.7)	33.3 (20.3 – 46.2)	33.1 (22.6 – 45.7)	0.8 (0.6 – 1.2)	0.8 (0.6 – 1.1)
Staten Island	30 (5.5)	26.4 (1.6 – 51.2)	13.5 (4.8 – 32.3)	0.7 (0.4 – 1.3)	0.6 (0.3 – 1.2)
Vaccination status					
Boosted	393 (70.9)	43.9 (35.8 – 51.9)	43.3 (35.6 – 51.3)	1.4 (1.0 – 2.2)	1.5 (1.0 – 2.2)
Fully vaccinated not boosted	58 (10.5)	30.4 (12.2 – 48.5)	33.4 (21.8 – 47.4)	-ref-	-ref-
Not vaccinated	103 (18.6)	33.8 (18.9 – 48.6)	24.7 (15.6 – 36.7)	1.1 (0.7 – 1.8)	1.1 (0.6 – 1.7)

(cont'd)

Prior COVID since March 2020					
Never	151 (27.3)	27.5 (14.2 – 40.9)	27.1 (17.9 – 38.8)	0.4 (0.3 – 0.6)	0.7 (0.6 – 0.9)
Once	139 (25.2)	66.0 (54.7 – 77.2)	62.6 (52.2 – 71.9)	-ref-	-ref-
More than once	114 (20.5)	48.7 (33.5 – 63.8)	45.8 (33.1 – 59.1)	0.7 (0.6 – 0.9)	0.9 (0.8 – 1.0)
Never tested positive but think they had COVID	150 (27.3)	24.0 (14.3 – 33.8)	21.8 (14.1 – 31.9)	0.4 (0.3 – 0.5)	0.7 (0.6 – 0.8)
Protection against severe disease					
Hybrid protection	314 (56.7)	47.5 (38.8 – 56.1)	47.9 (39.7 – 56.1)	-ref-	-ref-
Vaccine-induced protection only	137 (24.7)	29.9 (15.4 – 44.4)	29.1 (19.3 – 41.3)	0.6 (0.5 – 0.8)	0.6 (0.5 – 0.8)
Infection-induced protection only	89 (16.1)	38.4 (21.4 – 55.4)	27.4 (18.1 – 39.4)	0.8 (0.6 – 1.1)	0.7 (0.5 – 1.0)
No protection	14 (2.5)	4.6 (0.0 – 13.6)	3.7 (0.6 – 19.6)	0.1 (0.0 – 1.1)	0.2 (0.0 – 0.9)
Comorbidities					
Y	240 (43.3)	53.9 (43.4 – 64.5)	53.0 (42.7 – 63.1)	1.8 (1.4 – 2.2)	1.6 (1.3 – 1.9)
N	314 (56.7)	30.4 (22.5 – 38.4)	29.5 (22.8 – 37.2)	-ref-	-ref-
Any vulnerability*					
Y	355 (64.0)	47.0 (38.8 – 55.2)	47.7 (39.7 – 55.8)	1.6 (1.3 – 2.1)	2.0 (1.5 – 2.5)
N	199 (36.0)	29.2 (18.7 – 39.6)	32.8 (23.5 – 43.6)	-ref-	-ref-
Health insurance					
Yes	428 (77.2)	44.4 (37.0 – 51.8)	44.3 (37.1 – 51.7)	-ref-	-ref-
No	126 (22.8)	27.6 (14.7 – 40.2)	24.7 (15.1 – 37.9)	0.6 (0.5 – 0.8)	0.6 (0.4 – 0.8)
*Aged 65 or older OR ≥ 1 comorbidity OR unvaccinated					
**Direct standardized for the age and sex groupings in the 2020 US census					
***Model adjusted for sex and age					
****Estimates are adjusted for age only					

I have concerns with the paragraph that follows and appears to argue that this rapid survey approach can address biases related to differences in exposure and/or testing behaviors. As noted above, if a certain proportion of the population never tests for COVID-19 then they are less likely to identify their infection and report themselves positive on a survey. This seems highly possible/probable in a home setting given that many upper respiratory tract infections come and go without diagnosis. While studies with a TND design, cohorts with serial testing, and seroprevalence studies all have their own limitations they do attempt to address this testing issue in different ways, whereas the approach used here doesn't currently include any way of addressing this concern. Would consider removing the statement "Survey data such as ours can be used to correct VE estimates for these biases" or clarifying further.

Authors' response: Thank you for your comment. To be sure, our case definition allows for cases of symptomatic covid among individuals who do not test, but would not pick up asymptomatic infections among people who never test. Nonetheless, we maintain that including non-testers and assessing the presence of symptoms and known COVID-19 exposures during a surge reduces bias in prevalence estimates over those that are restricted to those who test. Our statement about the use of survey data to correct VE estimates is in terms of addressing selection bias as it relates to testing and health-seeking behaviors. One of the main limitations of the TND method for measuring VE is that those who are motivated to be tested and vaccinated are also more likely to access health care services than those who are not highly motivated. In this case, VE can be underestimated because vaccinated persons with positive tests would be over-represented. Surveys like ours, which ascertain cases without need to seek care (i.e., among

those untested [using probable case definition] and those who test at-home) can address this limitation.

Discussion Page 15: Survey data such as ours can be used to correct VE estimates for these biases by addressing selection bias as it relates to testing and health-seeking behaviors (i.e., those who are motivated to be tested and vaccinated are more likely to access health services).

Pg 12 - Since this manuscript was developed, there have now been a few articles showing a benefit of nirmatrelvir/ritonavir among vaccinated individuals (Effectiveness of nirmatrelvir-ritonavir against hospital admission: a matched cohort study in a large US healthcare system | medRxiv; Oral Nirmatrelvir and Ritonavir in Nonhospitalized Vaccinated Patients With Coronavirus Disease 2019 (COVID-19) | Clinical Infectious Diseases | Oxford Academic (oup.com))

Authors' response: Thank you for the citations. We have incorporated these references in the discussion, as follows:

Discussion, Page 17:

Our study suggests that awareness and uptake of nirmatrelvir/ritonavir (PaxlovidTM) was low among adults with SARS-CoV-2 infection in our study. Nirmatrelvir/ritonavir trials were conducted among unvaccinated individuals at high risk for hospitalization and death, and were shown to reduce the likelihood of these outcomes by ~90%.³ It is unclear how much added protection nirmatrelvir/ritonavir provides over and above that provided by vaccines/boosters. A study among vaccinated adults who received nirmatrelvir/ritonavir had reductions in ER visits, hospitalization, or death³¹. CDC recommends³² antivirals for individuals susceptible to severe COVID-19, regardless of vaccination status.

Regarding the statement on page 16 “But from an epidemiologic standpoint, these individuals should be reflected in prevalence estimates as they are actively infected” it is not clear what is meant by “actively infected,” but in most studies people who are immunocompetent no longer shed viable/culturable virus beyond 10-12 days, whereas the PCR test can remain positive for weeks after. (Comparison of Home Antigen Testing With RT-PCR and Viral Culture During the Course of SARS-CoV-2 Infection | Infectious Diseases | JAMA Internal Medicine | JAMA Network)

Authors' response: Thank you for your comment. We have clarified that it is positive PCR tests 14 days or more after diagnosis that could result in an inflated estimate of prevalence. But we feel that positive antigen tests should be counted in any prevalence estimate 14 days or more after diagnosis, as they are in fact prevalent infections that could result in onward spread. The sentence now reads as follows:

Discussion, Page 21:

But from an epidemiologic standpoint, individuals with positive antigen tests should be reflected in all prevalence estimates as they are actively infected that could result in onward spread.

Reviewer #3 (Remarks to the Author):

The work estimates crude prevalence rates of SARS-CoV-2 infections in the last two weeks using a survey-based approach in NYC. Based on integration of questions on risk factors, the authors further identify potential contributors to increased risk of infection including increasing household size and children under-18 (likely confounded with household size). Estimated rates

are then qualitatively compared to other data sets, including reported case data, hospitalizations, deaths, and SARS-CoV-2 RNA concentrations in wastewater. The authors estimate that there is a 30-fold higher estimate of infections in the prior two weeks than is reported in the official case count, and that the magnitude of the surge in BA.2 infections was significantly underestimated by alternative monitoring methods. The authors then extend their findings to suggest that immunity helped to prevent a surge in hospitalizations and deaths. The methodology provides insights into the extent to which survey-based data on infections may be useful for estimating crude-prevalence rates.

Authors' response: Thank you for your review and feedback. We examined whether the increased risk of infection among children under 18 was likely confounded by household size. Since household size is correlated with households with children, we estimated among households without children, the prevalence of SARS-CoV-2 in higher household sizes compared to households with size of 1. We found that an increase in household size did not increase risk of infection compared to households with 1 member. This finding suggests that presence of children in households increases risk of infection.

1. There is clear bias in survey-based metrics (non-response, social desirability), particularly given the low response rate. Although the authors acknowledge this, the manuscript would benefit from a more nuanced discussion of their impacts on the study conclusions. If the survey-based approach shows dramatically higher incidence, should the survey results be taken as the standard metric to which all other metrics are insufficient? I would argue that no, the true (unobservable) prevalence rate is likely between the upper and lower bounds. If so, what additional value does phone-based surveys provide?

Authors' response: Thank you for your comment. We have updated the discussion to include non-response bias and social desirability as biases that could affect the validity of our survey prevalence estimates. We agree with the reviewer's assessment that the point estimate of any study is subject to bias and is likely to be between the upper and lower bounds. We believe surveys like ours have an important value in that they contribute to the understanding of where the burden of infection stands during a surge across subpopulations in a current landscape where there is increasingly limited data that are not subject to testing bias. Even if the true estimate is closer to the lower bound, the burden is still higher than what is reflected in standard surveillance. We also argue that surveys provide an important snapshot of which vulnerable groups are least likely to test and those who have the highest burden of infection, since they can collect more data than is typically gathered as part of routine diagnostic testing activities.

Phone-based or even in-person household-based surveys, even if truly probability based, suffer from low response rates. Population-based surveys in the US have been seeing increasingly low response rates but as a recent paper by the Pew Research Center notes, phone-based surveys can still provide accurate data even with low response rates. Household surveys require intensive resources in following up on households to participate and can also have low response rates. For example, the REACT-1 survey that samples 180,000 households has a response rate of 12.1%. Even with a relatively higher response rate, the REACT-1 study authors note in their methods that non-response bias cannot be ruled out if those who participate are more or less likely to be infected. To assess the impact of non-response on study validity, studies that have examined non-response in surveys suggest that response rates are a poor indicator of non-response bias and data quality.

We have added the following language to the limitations section in the Discussion:

Discussion, Page 22: While non-response bias could affect the validity of our survey prevalence estimates, we believe surveys like ours have an important value in that they contribute to the understanding of where the burden of infection stands during a surge across subpopulations in a current landscape where there is increasingly limited data that are not subject to testing bias. Even if the true prevalence estimate of SARS-CoV2 is closer to the lower bound, the burden is still higher than what is reflected in standard surveillance. Cross-sectional prevalence surveys also provide an important snapshot of which vulnerable groups are least likely to test and those who have the highest burden of infection, since they can collect more data than is typically gathered as part of routine diagnostic testing activities.

2. A discussion of the proposed methods relative to other methods that complemented cross-sectional surveys with objective indicators (PCR-testing, serology) (see Stringhini et al. 2020 and 2021 in Geneva; Layton et al. 2022 in Oregon) is warranted, particularly on the relative reliability of the resulting estimates.

Authors' response: Thank you for these suggestions. We added a paragraph under the limitations of the study that addresses how surveys like ours can be enhanced if they are both routinely deployed and combined with biomarker data that would enable the: 1) capturing of valid estimates of prevalence and 2) detecting population-level immunity due to prior infection and vaccination. In addition, we also mention possible trade-offs that would need to be considered if also collecting biomarker data.

Discussion, Page 22: Survey-based approaches to measure population-representative prevalence estimates could be vastly enhanced if these surveys are strategically and routinely deployed and combined with biomarker indicators. Surveys that have incorporated PCR testing in prevalence estimation could validate the prevalence estimates based on self-report and asymptomatic cases that would otherwise be missed⁵⁰. Routinely deployed population-representative cross-sectional surveys are also better equipped to detect surges⁴³ and to triangulate with wastewater-based surveillance so that trends may also be compared. If combined with serological testing, routinely deployed seroprevalence surveys can provide time-series trends⁵¹ that can elucidate levels of population immunity due to prior infections and vaccinations⁵². However, it is also possible that the addition of biomarkers and related procedures could reduce participation and timeliness. A hybrid approach that includes biomarkers as a validation procedure could be used to correct self-reported data in the same survey or in similar surveys that do not use biomarkers.

3. The qualitative comparisons to other metrics is limited by the single point prevalence estimate available in the cross-sectional survey relative to the historical time series data of other metrics. All of the data sources are limited, as the authors note, but the time series data benefits from temporal (dynamic) comparisons. Given the relative ease and low-cost of phone-based surveys, such an approach could easily be extended to gather time series data that would be more comparable. See minor comments on limitations of the qualitative comparisons to wastewater.

Authors' response: We agree that repeated surveys to establish a trend would be of great value. This could be done by the CDC or individual state/city/county health departments, and would enhance comparisons of trends over time for data on wastewater concentrations, cases, hospitalizations, and deaths. We have added the following statement to the Discussion:

Discussion, Page 23:

Survey-based approaches to measure population-representative prevalence estimates could be vastly enhanced if these surveys are strategically and routinely deployed and combined with

biomarker indicators. Surveys that have incorporated PCR testing in prevalence estimation could validate the prevalence estimates based on self-report and asymptomatic cases that would otherwise be missed⁵⁰. Routinely deployed population-representative cross-sectional surveys are also better equipped to detect surges⁴³ and to triangulate with wastewater-based surveillance so that trends may also be compared. If combined with serological testing, routinely deployed seroprevalence surveys can provide time-series trends⁵¹ that can elucidate levels of population immunity due to prior infections and vaccinations⁵². However, it is also possible that the addition of biomarkers and related procedures could reduce participation and timeliness. A hybrid approach that includes biomarkers as a validation procedure could be used to correct self-reported data in the same survey or in similar surveys that do not use biomarkers.

Minor comments:

“intervals between surges”, which is this distinct from “population levels of immune protection”?

Authors’ response: Our intent was to convey that when there are longer intervals between surges, that hybrid protection may wane and leave some communities more susceptible to surges in hospitalizations and deaths.

Introduction, Page 6: ...,varying intervals between surges which in the case of longer intervals, protection may wane and leave some communities more susceptible to surges in hospitalizations and deaths,....

“22.1% of respondents”, having a sample size here would be useful.

Authors’ response: Thank you for your comment. We specified the sample size in the following statements.

Abstract, Page 2: An estimated 22.1% (95%CI 17.9%-26.2%) of 1,030 respondents had SARS-CoV-2 infection during the two-week study period, corresponding to ~1.5 million adults (95%CI 1.3-1.8 million).

Results, Page 10: An estimated 22.1% (95%CI 17.9%-26.2%) of 1,030 respondents had SARS-CoV-2 infection in the 14 days prior to the interview, corresponding to 1.5 million adults (95% CI 1.3-1.8 million) (Table 1).

“We plotted the mean per capita SARS-CoV-2 load (n1 copies per day per population)”. Capita and population are reported twice in this description. Load is typically reported as copies per day. This may be an error, or it may be a miscalculation. The visualization of mean per capita SARS-CoV-2 load assumes temporal alignment of disease dynamics with normal distributions, which is not likely accurate because there are likely spatially-resolved heterogeneity in timing of peaks and troughs. Have the authors considered visualizing a summed loading? The dynamics are likely similar, but this would be more mechanistic (“total load”).

Authors’ response: We are sorry that the labeling caused confusion. We used the exact wording provided by the New York State Department of Environmental Protection. We now describe this as “Per capita SARS-CoV-2 load”, and report it in units of N1 gene copies/capita. We have also updated the description of how wastewater sampling is performed to help better contextualize the data we present. Each of the 14 water resource recovery facilities (WRRFs) is intended to be sampled twice per week (it is not daily sampling), and sometimes it is only once per week. We took an average of the per capita SARS-CoV-2 viral load across the WRRFs for

samples that were collected on the same sample collection dates in a given week. While the reviewer is correct that averaging across WRRFs that serve different geographic areas assumes temporal alignment of disease dynamics, this is probably a reasonable assumption on a weekly basis for a small geographic area like NYC. To examine this, we looked at the per capita SARS-CoV-2 viral load by WRRF and the trends are fairly well-aligned (top figure below). We also examined the total load as opposed to the mean load as suggested by the reviewer and the trend did not change (middle figure below) in comparison to the mean (bottom figure below). Given this, we have chosen to report the mean per capita SARS-CoV-2 viral load, which we now report with confidence intervals, as suggested by the reviewer.

Revisions to methods

SARS-CoV-2 wastewater surveillance data, Page 9:

We analyzed publicly available data on SARS-CoV-2 concentrations in NYC wastewater through June 5, 2022, which is estimated based on influent samples from 14 water resource recovery facilities (WRRFs) in NYC covering wastewater of an estimated 8.2 million residents.²⁰ Specifically, WRRFs are sampled up to twice each week and per capita SARS-CoV-2 load (N1 copies per capita) is reported for each sample date.²⁰ We plotted the mean per capita SARS-CoV-2 load (N1 copies per capita) by sample date across all 14 WRRFs. Details on sampling and laboratory methods and measurement are available in the public use dataset documentation.²⁰

(cont'd)

Figure 1, why not include error bars to highlight variation and uncertainty, in line with recommendation for data presentation guidelines (clearly defined error bars).

Authors' response: Thank you for your comment. We have modified the Figure to include error bars.

Figure 1. Age- and sex-adjusted SARS-CoV-2 prevalence estimates among NYC adults by household size and presence of children in the household, April-May, 2022

Figure 4; Define Variant Era (50% or more samples based on clinical sample sequencing?) in the Figure caption.

Figure S1 has a strange formatting issue at 1000. Was this image overlaid with a line or otherwise formatted strangely? What is this ratio describing? The data in Figure 4 or the loads? A more clear description of the aggregation of data from the 14 WWTPs would be beneficial in this graphic.

Authors' response: We have updated the captions of all of the figures caption to include how the variant eras were defined. We have also added a description of the aggregation procedures for per capita SARS-CoV-2 viral load in Figure 4. We did not notice a strange formatting issue in our version of Figure S1. The new caption for Figure 4 now reads as follows:

Figure 4 caption:

Figure 4. Mean per capita SARS-CoV-2 concentrations from 14 water resource recovery facilities (WRRFs) in NYC covering wastewater of an estimated 8.2 million residents. WRRFs are sampled up to twice each week and per capita SARS-CoV-2 load (N1 copies per capita) is reported for each sample date. We plotted the mean per capita SARS-CoV-2 load by sample date across all 14 WRRFs. Variant eras for NYC were approximated /based on the timing of peaks and troughs in COVID-related hospitalizations in NYC as follows: 1 December 2021-1 March 2022 (BA.1) and 1 March 2022-6 June 2022 (BA.2).

“as many pandemic restrictions...” suggest providing citations for this.

Authors' response: Thank you for your comment. We have cited the restrictions that have been dropped in NYC as was announced on March 2, 2022. We included the following citation:

<https://www.nyc.gov/office-of-the-mayor/news/108-22/as-covid-cases-plummet-vaccination-rates-reach-new-heights-mayor-adams-next-phase-of/#/0>

“Despite our estimate...” Suggest the authors revise this section. The authors did not conduct a survey during BA.1, and so can not draw conclusions about the relative number of people infected during BA.1 compared to their estimated 1.5M during 2 weeks of BA.2. The current study does not detect a “surge”, it only estimates a crude prevalence. Although they find 1.5 million people were infected in the last two weeks, the relatively high wastewater loads observed during this period (Figure 4) could explain this, but in the absence of a time series of survey-based methods, it is impossible to observe differences in survey to wastewater concentration ratios.

Authors' response: We have removed the first sentence of that section for reasons mentioned by the reviewer.

“If this is true, “, it appears from the figures that the wastewater signal performed at least as well in observing the surge as other time series indicators (reported cases, hospitalizations, and deaths). See prior comment on the inability of a cross-sectional survey to detect a surge.

Authors' response: We have removed the last two sentences of this section for the reasons mentioned by the reviewer.

“Indeed, a very large... “ What are the clinical samples discussed here? Although the authors make a strong and important point about the potential of prior infection or vaccination to attenuate the ratio between true (unobservable) infections and wastewater concentrations, as written the point of this section is not so clear.

Authors' response: Thank you for your comment. We clarified in this section that the study used clinical samples from over 380,000 individuals in a cohort who had active primary, reinfection, or breakthrough infections.

Discussion, Page 18:

Indeed, a very large pre-Omicron study from Qatar compared cycle threshold (Ct) values of over 380,000 individuals with active primary, reinfection, or breakthrough infection, adjusting for sex, age, reason for testing and calendar week of testing.

“in a recent EHR”, define HER.

Authors' response: Thank you. We have included the definition EHR, electronic health record, to clarify.

Discussion, Page 19:

“However, in a recent electronic health record (EHR) based analysis...”

“Our survey estimates of... “Non-response bias discussed here may play a role, acquiescence or social desirability biases may also artificially inflate estimates. See major comments.

Authors' response: Thank you for your comment. We have modified this section. See modified text below:

Discussion, Page 21:

Our survey estimates of SARS-CoV-2 prevalence and provider testing are subject to selection and information biases. In terms of selection bias, our survey estimates may be biased due to non-response if those who responded differ from non-respondents. We could not correct for this bias; however, characteristics of survey respondents did not differ substantially from that of the adult NYC population (Appendix 1). In addition, our estimates could be inflated if those who both tested for SARS-CoV-2 infection and tested positive were more likely to participate in the survey than those who did not. While potential survey participants were not aware of the survey content before deciding to participate, it may be that those who were positive were more likely to complete the survey. . Our survey response rate was low (1.2%) given that all sampling was done through phone-based random digit dialing (see Appendix 1). While phone-based surveys have increasingly been subject to low response rates over time⁴⁶, studies have shown that rates are a poor indicator of non-response bias and data quality⁴⁷. In terms of information bias, it is also possible that participants inadvertently recalled and reported positive tests that were beyond the 14-day study period (recall bias). Lastly, some people test multiple times with providers after their initial positive test⁴⁰, and subsequently, many can expect positive PCR and antigen test results for 10 or more days.^{44,45} This could have caused some people who were diagnosed prior to the study period to have positive tests during the study period which could have inflated our prevalence estimates relative to official case counts.

“From an epidemiological standpoint, “ Although it is true these people could be included as active shedders in estimates within the scope of infectious disease modeling, these authors otherwise do not include (or discuss) these people within the scope of other metrics used. In comparing authors results to other data sets, this should be held consistent or recognized as a clear limitation of the study’s ability to compare.

Authors' response: We have clarified that it is positive PCR tests 14 days or more after diagnosis that could result in an inflated estimate of prevalence. But we feel that positive antigen tests should be counted in any prevalence estimate 14 days or more after diagnosis, as they are in fact prevalent infections that could result in onward spread. The sentence now reads as follows:

Discussion, Page 22:

But from an epidemiologic standpoint, individuals with positive antigen tests should be reflected in all prevalence estimates as they are actively infected that could result in onward spread.

“was underestimated by” it is generally well-accepted (and not novel) that official case counts represent a portion of true (unobserved cases). For wastewater, the discussion highlighting the insufficiencies is flawed (see prior comment), and the authors do not provide a basis for estimating incidence based on wastewater concentrations. Suggest the authors revisit this statement.

Authors' response: We have changed the first sentence of our conclusions to say “Our study attempted to quantify the extent to which the magnitude of NYC’s BA.2/BA.2.12.1 surge was underestimated by...”

“Limiting the surge’s impact” the authors attribute lack of hospitalizations and deaths to prior infection, but this does not follow directly on the research.

Authors’ response: Review with Denis – suggest removing “limiting the surge’s impact” or rewording since it suggests causal impact (and can’t be determined with cross-sectional survey). We have revised as follows:

Even though many individuals vulnerable to a severe outcome were infected and did not use rapid antivirals, most had a high degree of protection against a severe outcome through vaccination and boosters, overlaid by a recent history of prior BA.1 infection, which may have limited the surge’s impact on hospitalizations and deaths.

REVIEWERS' COMMENTS:

Reviewer #3 (Remarks to the Author):

The authors have responded thoughtfully and thoroughly to all previous reviewer 3 comments. One final comment remains, and one minor comment:

The Conclusion states

"... underestimated by official case counts and wastewater surveillance...". The authors should strongly consider rewording this, as it suggests deficiencies in official case counts and wastewater surveillance. As previously discussed, it is well known official case counts represent a portion of cases; further: wastewater can currently only inform trends, not prevalence rates. As such, the discussion on underestimation of wastewater is flawed. My understanding is that the authors conclude this by looking at differences in shedding between BA.1 and BA.2, and suggest because BA.2 was lower than BA.1, wastewater must not be underestimating. I think this is flawed, because the authors do not have survey data for the BA.1 surge (which may have shown higher prevalence rates than observed for BA.2, in line with trends in wastewater).

Suggest authors revisit this conclusion, and shift the tone, highlighting the power of surveys a rapidly deployable and informative approach complementary to alternative methods that may provide more insights.

Minor comments:

Reviewer 3, final comment response appears to be an internal note from authors amongst themselves. Suggest revisiting this comment. The point remains that this last sentence (line 575: declaring the reduction in ratio of hospitalizations/deaths to cases as a result of prior infection) is outside of the scope of this work. Suggest removing, or discussing more in line with the work (survey results to estimate cases could be used in the future to understand this ratio, and potentially inform severity of variants, for example).

REVIEWERS' COMMENTS:

Reviewer #3 (Remarks to the Author):

The authors have responded thoughtfully and thoroughly to all previous reviewer 3 comments. One final comment remains, and one minor comment:

The Conclusion states

"... underestimated by official case counts and wastewater surveillance...". The authors should strongly consider rewording this, as it suggests deficiencies in official case counts and wastewater surveillance. As previously discussed, it is well known official case counts represent a portion of cases; further: wastewater can currently only inform trends, not prevalence rates.

As such, the discussion on underestimation of wastewater is flawed. My understanding is that the authors conclude this by looking at differences in shedding between BA.1 and BA.2, and suggest because BA.2 was lower than BA.1, wastewater must not be underestimating. I think this is flawed, because the authors do not have survey data for the BA.1 surge (which may have shown higher prevalence rates than observed for BA.2, in line with trends in wastewater).

Suggest authors revisit this conclusion, and shift the tone, highlighting the power of surveys a rapidly deployable and informative approach complementary to alternative methods that may provide more insights.

Authors response: *Thank you for your comment. As suggested, we have removed the reference to the underestimation of wastewater surveillance. We have revised the text in the conclusion as follows:*

Conclusion, Page 24: *Our study characterized and quantified the extent to which the magnitude of NYC's BA.2/BA.2.12.1 surge was underestimated by official case counts, due to a combination of exclusive at-home testing, not testing at all, incomplete provider/laboratory reporting. This underestimation of burden likely also occurred in other U.S. jurisdictions, and by extension, the national SARS-CoV-2 surveillance system. Even though many individuals vulnerable to a severe outcome were infected and did not use rapid antivirals, most also had a high degree of protection against a severe outcome through vaccination and boosters, overlaid by a recent history of prior BA.1 infection.*

At the outset of surges, given the of how they will impact severe outcomes among those who remain susceptible, a shift in approach to public health surveillance for SARS-CoV-2 is needed. More routine and timely quantification of SARS-CoV-2 infection burden can allow for the monitoring of the ratio of infections to hospitalizations and deaths, and thus characterize the severity of surges, including those due to novel variants. Our findings demonstrate the utility of population-representative surveys as an important surveillance tool to go alongside, and

triangulate with, passive case reporting, genomic surveillance, and wastewater surveillance at uncertain and evolving stages of the U.S. pandemic.

Minor comments:

Reviewer 3, final comment response appears to be an internal note from authors amongst themselves. Suggest revisiting this comment. The point remains that this last sentence (line 575: declaring the reduction in ratio of hospitalizations/deaths to cases as a result of prior infection) is outside of the scope of this work. Suggest removing, or discussing more in line with the work (survey results to estimate cases could be used in the future to understand this ratio, and potentially inform severity of variants, for example).

Authors response: *Thank you for your comment and apologies for the oversight. As suggested, we have removed the reference about the reduction in ratio of hospitalizations and deaths. We have revised the text in the conclusion as detailed in the response to the prior comment.:*